# Integrated Forward–Inverse Network for Physics-Guided Image Reconstruction

## Abstract

Inverse modeling plays a central role across computational optical imaging problems, including microscopy, imaging through scattering media, and lensless cameras, where the forward model often manifests as a severe blur. Discrepancies between the model and the actual imaging process further aggravate the ill-posed nature of the inverse problem. Physics-enabled methods that integrate analytical forward models with data-driven networks have been explored, but most incorporate priors only in a one-sided manner—either operating purely in the measurement space or only after inversion—thereby discarding complementary cues and reducing robustness to calibration errors. Here, we propose the Integrated Forward–Inverse Network (IFIN), a physics-guided deep neural network that interleaves differentiable forward operators with learnable inverse modules at every stage of the hierarchy. This design preserves physical consistency while shaping richer feature representations by jointly leveraging information from both measurement and image domains. A physics-guided kernel adaptation further compensates for inaccurate or unavailable PSF calibration, dynamically refining the kernel for blind deconvolution under system constraints. IFIN is especially effective when measurements are severely blurred by large point-spread functions, where conventional CNN-based inversion is limited by local receptive fields and underutilizes the measurement signal. On challenging lensless imaging benchmarks—including our newly introduced dataset, IFIN achieves state-of-the-art reconstruction quality and improved robustness under noise and model mismatch.

## 1 Introduction

Modern optical imaging systems—ranging from compact lensless cameras with coded apertures to advanced microscopes with engineered point-spread functions (PSFs)—are increasingly designed with complex forward models. These systems often operate in regimes where the PSF is intentionally or unavoidably broadened by scattering, diffraction, or designed optical coding, while producing significantly blurred measurements. Such designs unlock diverse imaging capabilities (Sahoo et al., 2017; Satat et al., 2017; Antipa et al., 2017; 2019; Baek et al., 2022) that transcend the limits of conventional optics, yet they also introduce substantial challenges for reconstruction. In particular, the resulting measurements frequently violate the stationarity assumptions underlying standard inverse pipelines (Kuo et al., 2020; Cai et al., 2024), while hardware imperfections and residual modulations further complicate the model. As optical platforms continue to shrink and diversify, the model mismatch between the assumed forward model and the actual device increases, making accurate and robust image restoration a central difficulty.

A wide spectrum of approaches has been explored for image reconstruction under complex forward models. Classical, analytically derived inverse mappings (Wiener, 1964) and model-based optimization (Richardson, 1972; Lucy, 1974; Boyd et al., 2011) built on well-defined priors offer stable and physically valid results, but the methods are often computationally expensive, sensitive to calibration errors, and unreliable under model mismatch. With advances in deep learning, data-driven methods (Anonymous, 2020; Pan et al., 2022) have enabled end-to-end mappings from measurements to target scenes. While such models provide fast inference and can be trained directly on task metrics, they may not explicitly encode the underlying system physics, which can reduce accuracy and robustness under out-of-distribution conditions and occasionally yield hallucinations.

In response, hybrid methods that embed the physical forward model within a learning framework have emerged, improving efficiency and grounding predictions in the physical model while leveraging data-driven components to capture priors that are difficult to specify analytically. Yet in practice, the methods typically integrate physics only in a one-sided manner—for instance, by applying a learned denoiser before or after pure physical reconstruction—either closed-form or optimization-based—with or without learnable parameters (Monakhova et al., 2019; Khan et al., 2020; Yanny et al., 2022; Kingshott et al., 2022; Poudel & Nakarmi, 2024; Lee et al., 2023a; Bezzam et al., 2025), incorporating the forward model into the loss term (Ulyanov et al., 2018; Monakhova et al., 2021), or embedding an inverse mapping into a single network layer (Dong et al., 2021; Li et al., 2023; Bai et al., 2025).

Such strategies incorporate physics in a one-sided manner: they either operate purely in the measurement space or rely only on an already inverted image, and once a deconvolved estimate is produced the raw measurements are typically discarded. Under severe blur, this is problematic: once the pipeline collapses everything into a single reconstructed image, fine details that are difficult to infer after inversion cannot be reliably recovered, whereas operating only on raw measurements leaves purely data-driven layers without enough structure to disentangle the underlying scene. To address this, we seek architectures that maintain raw measurements and image-domain representations in tandem and enable the forward and inverse operators to act as paired, multi-scale modules that guide feature evolution under the system's physics, including shift-variant and mismatched PSFs.

To this end, we introduce **IFIN**, a unified encoder-decoder architecture that embeds differentiable forward operators and learnable inverse modules at every stage of the hierarchy. This design not only preserves physical consistency but also shapes richer feature representations by jointly leveraging measurement- and image-domain information. In addition, a physics-guided kernel adaptation compensates for imperfect or unknown PSF calibration; when direct PSF measurement is infeasible, the kernel is dynamically refined for blind deconvolution under the constraints of system physics. Our main contributions are as follows:

- **Integrated forward–inverse:** An encoder-decoder design where differentiable forward operators and learnable inverse modules are integrated at every level, maintaining physics consistency while shaping richer feature representations.
- **Learnable spatially variant modeling:** A parameterization that captures lateral shift- and channel-dependent variations, regularized for stability to enable accurate recovery in complex systems, with a learnable kernel representation jointly optimized with reconstruction, allowing blind deconvolution when PSF calibration is inaccurate or unavailable.

This explicit integration of forward and inverse processes is particularly effective in regimes where the measurements are heavily degraded, such as when the imaging system produces severely blurred data due to large PSFs. Data-driven inversion with conventional CNNs often struggles to capture the broader correlations required in these regimes, leaving much of the measurement signal under-utilized. In contrast, our design propagates information through both the measurement and image domains at every stage, performing physics-guided inversion while simultaneously learning representations that capture variations difficult to model analytically. To illustrate the benefits of proposed framework, we focus on lensless imaging as a representative case, where our approach demonstrates superior performance compared to prior methods.

## 2 RELATED WORK

### 2.1 LENSLESS IMAGING

Lensless cameras replace conventional lenses with thin optical elements such as coded aperture (Asif et al., 2016), transmissive diffusers (Antipa et al., 2017), and engineered phase masks (Lee et al., 2023b). As a result, diffuser- or mask-induced PSFs are large and highly structured, often encoding wide spatial neighborhoods—up to the entire scene—onto the sensor. This encoding eliminates the need for bulky optics but necessitates computational reconstruction to recover interpretable images from the raw measurements.

Beyond simple image recovery, the same computational framework also unlocks a wide range of modalities, including depth estimation (Antipa et al., 2017; Bagadthey et al., 2022), hyperspectral

imaging (Sahoo et al., 2017; Monakhova et al., 2020), polarization analysis (Baek et al., 2022), single-shot ultrafast video capture via rolling-shutter coding (Antipa et al., 2019), and privacy-preserving imaging based on the expressive representations of lensless measurements (Satat et al., 2017; Henry et al., 2023). These capabilities, coupled with the ultra-compact and lightweight architecture, make lensless cameras particularly appealing for applications in embedded vision where size, cost, and its unique imaging functionalities are critical (Kim et al., 2024; Ge et al., 2024; Xiangjun & Yue, 2025).

Image reconstruction becomes particularly challenging when the optical system produces strongly spread measurements, often modeled as 2-D or 3-D convolutions with large kernels. In such cases, extended PSFs distribute scene information broadly across the sensor, leading to loss of spatial detail and strong overlap between measurements, which makes inversion ill-posed. Similar challenges arise in a range of computational imaging settings, from conventional cameras under severe aberrations or motion blur to advanced imaging tasks such as imaging through scattering media (Yoon et al., 2020), non-line-of-sight imaging (Faccio et al., 2020), coherent diffractive imaging (Miao et al., 2015), and advanced microscopy techniques with designed PSFs (Pavani et al., 2009).

We begin with a baseline shift-invariant model, which assumes that the system response is identical across all spatial locations. Under this assumption, the measurement is expressed as a 2-D convolution between the scene and a position-independent PSF:

$$y[i,j] \;=\; \sum_{k,\ell} h[i-k,\, j-\ell]\, x[k,\ell] \;+\; \eta[i,j], \tag{1}$$

where $x, y \in \mathbb{R}^{H \times W}$ denote the scene irradiance and the captured measurement, $h[\cdot, \cdot]$ is a position-independent PSF, and $\eta$ models additive noise.

## 2.2 Image Restoration

Given such forward models, image recovery in lensless cameras is carried out through computational inversion, often posed as deconvolution. This places lensless reconstruction in the same category of problems as deblurring in conventional cameras, where severe optical blur can arise from optical aberrations, motion, or atmospheric turbulence. A long line of approaches has been explored for this task. Classical methods—including Wiener deconvolution (Wiener, 1964), Richardson-Lucy (Richardson, 1972; Lucy, 1974) provide physically grounded solutions, but their performance quickly deteriorates under noise and kernel mis-specification. In practice, non-differentiable elements in the forward model (e.g., cropping or truncation) together with priors such as total variation (TV) regularization motivate the use of optimization frameworks like the Alternating Direction Method of Multipliers (ADMM) (Boyd et al., 2011), which decouple data fidelity and regularization terms to enable tractable iterative solvers (Antipa et al., 2017).

In parallel, substantial efforts address non-uniform blur using PSF fields—either calibrated or estimated (Robbins & Huang, 1972; Denis et al., 2015; Yeo et al., 2025), trading off additional calibration effort and computational cost. More recently, deep learning methods—including Convolutional Neural Network (CNN)-based (Ronneberger et al., 2015; Zhang et al., 2018b; Anonymous, 2020; Chen et al., 2022) and vision-transformer (ViT) architectures (Dosovitskiy et al., 2020; Pan et al., 2022)—have emerged as powerful data-driven approaches, learning end-to-end mappings from measurements to images and often achieving state-of-the-art restoration quality, yet they remain prone to hallucinations and limited generalization under diverse real-world degradations.

Building on the limitations of purely data-driven approaches, a growing body of work has incorporated explicit physical priors into neural reconstruction pipelines, rather than relying solely on a fully data-driven mapping from severely degraded measurements. One prominent direction unrolls classical optimization, embedding the forward operator directly into iterative updates: Monakhova et al. (2019), Kingshott et al. (2022), Poudel & Nakarmi (2024) and Bezzam et al. (2025) augment unrolled iterations with a neural denoiser, while Kingshott et al. (2022) adopts a primal–dual unrolling that jointly learns forward and adjoint operators. Forward-model constraints are also used for measurement consistency in unsupervised training, where neural priors alone can guide reconstructions without ground-truth supervision (Ulyanov et al., 2018; Wang et al., 2020; Monakhova et al., 2021). Another approach employs feed-forward hybrid architectures, where a physical inversion stage is followed by a learned refinement network (Khan et al., 2020; Yanny et al., 2022). A

related line of work performs deconvolution in feature space, embedding convolutional inversion within multi-scale skip connections to improve fidelity and robustness (Dong et al., 2021; Li et al., 2023; Bai et al., 2025). These hybrid methods demonstrate the benefits of combining physics and learning, but they typically operate within a single feature hierarchy and treat the acquired measurement y as a fixed input that is not updated once features are extracted. In contrast, IFIN maintains two coupled measurement and image streams and leverages multi-scale forward–inverse integration, as detailed in Section 4.

## 3 PRELIMINARIES

### 3.1 DEGENERATED FEATURE REPRESENTATIONS

We consider a linear forward model

$$y_\beta = H_\beta x + \eta, \tag{2}$$

where $x \in \mathbb{R}^n$ denotes the scene, $y_\beta \in \mathbb{R}^m$ the measurement, $H_\beta \in \mathbb{R}^{m \times n}$ a blur operator parameterized by complexity $\beta$, and $\eta$ sensor noise. As $\beta$ increases, the PSF grows larger and mixes information across increasingly distant pixels. Below we summarize four fundamental issues that arise when restoring severely blurred measurements with neural networks.

**(1) Effective-rank collapse.** The singular values of $H_\beta$ characterize how information is transferred from the scene to the measurement. Writing the condition number as

$$\kappa(H_\beta) = \frac{\sigma_{\max}(H_\beta)}{\sigma_{\min}(H_\beta)}, \tag{3}$$

severe blur corresponds to the regime in which $\sigma_{\min}(H_\beta)$ shrinks toward zero while $\sigma_{\max}(H_\beta)$ remains bounded, so that $\kappa(H_\beta)$ grows rapidly with $\beta$ and the inverse problem becomes increasingly ill-conditioned. This ill-conditioning is accompanied by a reduction in the effective rank of $H_\beta$. For a noise-dependent threshold $\tau > 0$, we define

$$\text{rank}_\tau(H_\beta) := \#\big\{ \sigma_i(H_\beta) : \sigma_i(H_\beta) > \tau \big\}, \tag{4}$$

and observe that as the kernel becomes larger and more diffuse, more singular values fall below $\tau$:

$$\frac{\partial}{\partial \beta} \text{rank}_\tau(H_\beta) \leq 0. \tag{5}$$

Hence $y_\beta$ lies in a progressively lower-dimensional subspace, and many scene components are mapped either to near-zero responses or to directions indistinguishable from noise (Bertero et al., 2021).

**(2) Purely neural restorers struggle with large-kernel blur.** Severe large-kernel blur mixes contributions from widely separated pixels into each measurement entry, while standard CNNs and ViTs process $y_\beta$ using finite receptive fields or windowed self-attention (Luo et al., 2016; Liu et al., 2021). When the blur kernel exceeds the effective receptive field or window size of a layer, each local feature receives a mixture of contributions from regions that cannot be jointly observed or disambiguated. From the layer's perspective, these out-of-field contributions behave as structured noise: they perturb local measurements without being explainable within the layer's spatial support. As a result, the difficulty introduced by the $H_\beta$ is further amplified by the mismatch between the blur footprint and the network's locality.

**(3) Inverse-induced information loss in one-sided reconstruction pipelines.** Beyond the information loss created by the forward operator $H_\beta$ itself, a second source of degeneration arises from how the inverse problem is solved. For a given measurement $y_\beta$, the set of scenes consistent with the forward model,

$$\mathcal{S}(y_\beta) = \big\{ x \in \mathbb{R}^n : H_\beta x \approx y_\beta \big\}, \tag{6}$$

is high-dimensional when $H_\beta$ is ill-conditioned: high-frequency variations are only weakly constrained by $y_\beta$ and produce indistinguishable measurements. Classical regularized optimization or deconvolution, as well as modern learned inverses, resolve this ambiguity by selecting a single estimate $\hat{x} = G(y_\beta)$ that favors smooth or low-complexity solutions, effectively suppressing such

weakly constrained directions (Bertero et al., 2021; Chen & Davies, 2020). One-sided physics–NN pipelines that apply $G$ and then feed only $\hat{x}$ into a network (Monakhova et al., 2019; Khan et al., 2020; Kingshott et al., 2022; Yanny et al., 2022; Poudel & Nakarmi, 2024; Bezzam et al., 2025) therefore incur an inverse-level loss of information: once $\hat{x}$ is fixed, the reconstruction network no longer has access to measurement-domain residuals indicating how $\hat{x}$ fails to explain $y_\beta$ along these weakly constrained high-frequency directions, and any remaining fine-scale detail is determined almost entirely by learned priors rather than by explicit consistency with the measurements.

**(4) Embedding physics inside networks is insufficient without preserving measurements.**
Even when inversion is embedded within neural layers (Dong et al., 2021; Li et al., 2023; Bai et al., 2025), the injected measurements often collapse into degenerate feature representations as they propagate through bottlenecked encoders. Let $f_\theta$ denote such an encoder that internally applies physics priors and produces

$$z_\beta = f_\theta(y_\beta). \tag{7}$$

Since $f_\theta$ is typically deterministic and dimension-reducing (due to downsampling, channel compression, activations, and normalization), it is non-invertible. By the data-processing inequality (Cover, 1999), any representation $z_\beta$ satisfies

$$I(x; z_\beta) \;\leq\; I(x; y_\beta), \tag{8}$$

and in practice the inequality is strict. Moreover, as $\beta$ increases and $\mathrm{rank}_\tau(H_\beta)$ decreases, the input $y_\beta$ becomes low-rank, causing a larger fraction of its components to fall below the effective thresholds of the encoder. Consequently, simply inserting inversion inside a conventional network is not sufficient: unless the architecture is explicitly designed to preserve and propagate the measurement information across layers, it remains vulnerable to the combined effects of ill-conditioned low-rank forward models and representation bottlenecks.

### 3.2 MODEL MISMATCH IN OPTICAL IMAGING

In practice, most imaging systems are not truly shift-invariant, even though their kernels are often modeled as convolutions. Off-axis aberrations, depth-dependent propagation, field-dependent magnification, vignetting or pupil clipping, and sensor truncation all make the effective system response depend on spatial location (Booth, 2014; Thiébaut et al., 2016; Antipa et al., 2017). This is especially pronounced for phase or coded masks with high effective numerical aperture: resolution improves on-axis, but aberration-induced shift variance grows with field angle. As a result, the PSF $h_{i,j}$ widens, skews, or changes phase structure across the field, necessitating a spatially varying model. A more general shift-variant model accounts for this effect by allowing the PSF to vary with the output coordinates:

$$y[i,j] \;=\; \sum_{k,\ell} h_{i,j}[k,\ell]\, x[k,\ell] \;+\; \eta[i,j], \tag{9}$$

where $h_{i,j}[k,\ell]$ is a location-dependent PSF at pixel position $[k,\ell]$. This formulation no longer reduces to a simple 2-D convolution, but instead to a large, spatially varying linear operator, which increases computational and memory demands for precise inversion. Sensor cropping (finite field-of-view truncation) and measurement noise further complicate inversion and increase the ill-posedness and susceptibility to calibration errors.

## 4 METHOD

The proposed IFIN adopts an encoder–decoder backbone equipped with Integrated Forward–Inverse Blocks (IFIBs) at every scale (Fig. 1). The encoder progressively downsamples the input measurement to capture large-scale blur effects and coarse scene structure, while the decoder upsamples features back to the native resolution to recover fine details. At each resolution, an IFIB couples a Forward System Operator (FSO), which maps image-domain features to the measurement domain, with an Inverse System Operator (ISO), which restores image-domain features from the measurement. Across scales, both operators are conditioned on and jointly refine a learnable PSF field, enforcing forward–inverse consistency while preserving raw measurement information throughout the network. At the input stage, a coarse inverse estimate is obtained by applying the ISO to the measurement. The pair *(measurement, coarse reconstruction)* is then propagated as two coupled streams

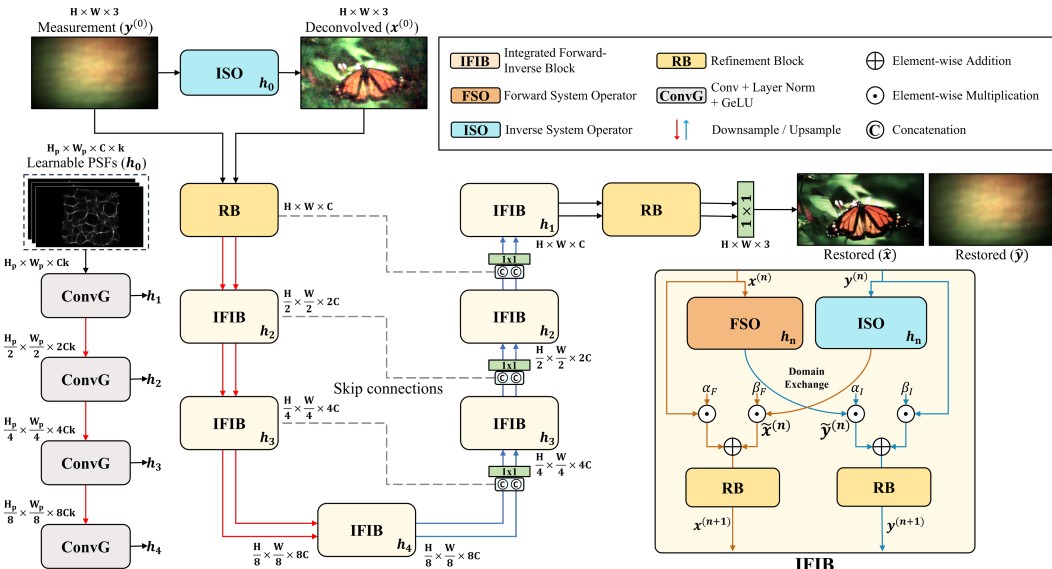

Figure 1: Overall architecture of IFIN. The network follows an encoder–decoder structure, where integrated Forward–Inverse Blocks (IFIBs) are inserted at each scale to jointly apply the Forward System Operator (FSO) and Inverse System Operator (ISO). A shared learnable PSF field guides both operators, ensuring forward–inverse consistency across scales.

through the encoder–decoder hierarchy. Within each IFIB, the FSO and ISO exchange features bidirectionally, so that measurement-domain consistency and image-domain fidelity are enforced jointly rather than in a one-sided manner.

## 4.1 LEARNABLE PSF FIELD

IFIN incorporates a learnable PSF representation that provides explicit system awareness to both FSO and ISO. The PSF field is parameterized as $k=s^2$ kernels covering local regions of the image. In case of $s=1$ (i.e., $k=1$), the PSF field reduces to a single global kernel. Kernels can be initialized from calibrated measurements, a single reference PSF, or random patterns, and are optimized jointly with the network.

A compact PSF encoder maps the field to multi-scale embeddings $\{h_n\}$ that condition all IFIBs throughout the network. Each embedding $h_n$ is shared between the encoder and decoder IFIBs at the same resolution level, so that the two streams at scale $n$ are constrained by a common system description. These embeddings are injected into both the FSO (image→measurement) and the ISO (measurement→image) at the corresponding resolution, maintaining physical consistency across the hierarchy. Because the same PSF field must simultaneously explain how image-domain features generate measurements (via the FSO) and how measurements are inverted back to sharp images (via the ISO), the PSF parameters are constrained by complementary supervision in both domains. This bidirectional usage yields stronger identifiability of the PSF field, discouraging degenerate kernel solutions and improving blind PSF estimation under severe blur (see Appendix A.5 for ablations on the PSF design).

## 4.2 INTEGRATED FORWARD–INVERSE BLOCK (IFIB)

The IFIB is the fundamental unit of IFIN, designed to couple forward and inverse imaging processes at each scale. Each IFIB consists of two parallel operators: (1) a Forward System Operator (FSO), and (2) an Inverse System Operator (ISO), as illustrated in Fig. 2. Both operators are fundamentally tied to the target system's forward and inverse physics. In practice, these operators can be configured as purely shift-invariant when degradations are approximately uniform, or as spatially varying to handle more complex degradations, allowing IFIN to balance efficiency and fidelity.

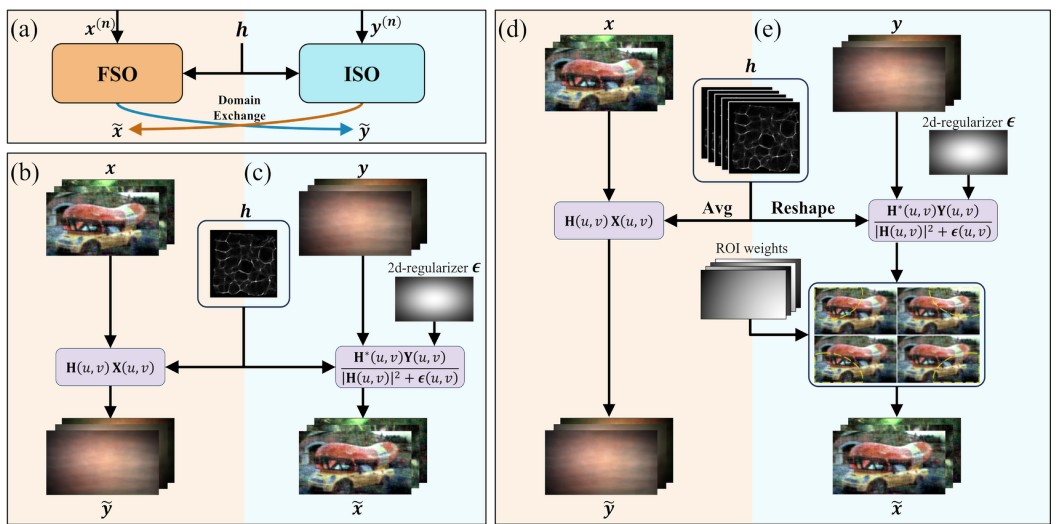

Figure 2: (a) Schematic of the forward–inverse pairing in the IFIB. (b) Shift-invariant (SI) FSO: 2-D convolution with a single PSF. (c) SI ISO: Wiener-like deconvolution in the frequency domain. (d) Shift-variant (SV) FSO: 2-D convolution with an average of PSF field. (e) SV ISO: region-wise deconvolution blended by learnable ROI maps.

### 4.2.1 FORWARD SYSTEM OPERATOR (FSO)

The FSO simulates how the current image-domain estimate $\hat{x}$ would be formed by the imaging system. By default we use 2-D linear convolution with zero padding via a single point-spread function $h$:

$$\tilde{y}[i,j] \;=\; (\hat{x} * h)[i,j]. \tag{10}$$

In the spatially varying case, we can model the FSO as a fully shift-variant convolution by tiling $\hat{x}$ and applying local PSFs with normalized overlap-add reassembly (see Appendix A.9). For computational efficiency in IFIN, we use a single-convolution surrogate with the averaged PSF $h_{\text{eff}} = \frac{1}{k}\sum_{r=1}^{k} h_r$.

### 4.2.2 INVERSE SYSTEM OPERATOR (ISO)

The ISO restores a sharp estimate from the degraded measurement via Wiener-like deconvolution with a learnable frequency-dependent regularizer. For PSF $h$, letting $Y(u,v) = \mathcal{F}\{W \cdot P_{rp}y\}(u,v)$ and $H(u,v) = \mathcal{F}\{h\}(u,v)$, where $\mathcal{F}\{\cdot\}$ denotes the Fourier transform, $P_{rp}$ is replicate padding, and $W$ is a mild Gaussian window used to mitigate wrap-around artifacts during deconvolution (Khan et al., 2020), we compute:

$$\widehat{X}(u,v) \;=\; \frac{H^*(u,v)}{|H(u,v)|^2 + \epsilon(u,v)}\, Y(u,v), \qquad \epsilon(u,v) > 0, \tag{11}$$

and set $\hat{x} = \mathcal{F}^{-1}\{\widehat{X}\}$. Here, $\epsilon(u,v)$ is a 2-D learnable parameterization predicted and refined during training, non-negativity enforced by a ReLU. By learning to estimate the distribution of noise from data, $\epsilon$ functions as an optimal frequency-selective prior that adapts to system noise.

In the spatially varying case, we perform region-wise deconvolution with distinct PSFs and blend the partial reconstructions:

$$Y(u,v) \;=\; \mathcal{F}\{W \cdot P_{rp}(y)\}(u,v), \qquad H_r(u,v) = \mathcal{F}\{h_r\}(u,v), \tag{12}$$

$$\widehat{X}_r(u,v) \;=\; \frac{H_r^*(u,v)}{|H_r(u,v)|^2 + \epsilon_r(u,v)}\, Y(u,v), \qquad \epsilon_r(u,v) > 0, \tag{13}$$

$$\hat{x}[i,j] = \sum_{r=1}^{k} w_r[i,j] \, \mathcal{F}^{-1}\{\widehat{X}_r\}[i,j], \tag{14}$$

where $\{w_r\}_{r=1}^{k}$ are learnable region-of-interest (ROI) maps, allowing the model to optimize the spatial support of each PSF over regions of varying extent and location. We initialize the ROI maps from Gaussian kernels $\{g_r\}_{r=1}^{k}$ centered at $\{p_r\}$, where $\{p_r\}$ are the centers of an $s \times s$ grid partitioning the input measurement.

$$g_r[i,j] = \exp\Big(-\frac{\|(i,j)-p_r\|_2^2}{2\sigma_r^2}\Big), \qquad w_r[i,j] = \frac{g_r[i,j]}{\sum_{q=1}^{k} g_q[i,j]}, \qquad \sum_{r=1}^{k} w_r[i,j] = 1 \ \forall(i,j). \tag{15}$$

This construction explicitly recovers different spatial neighborhoods using region-specific PSFs and priors, which is critical under spatially varying blur.

### 4.2.3 Integrated Forward–Inverse

The hallmark of IFIB is the bidirectional exchange between FSO and ISO. Each operator contributes complementary information to the other: the FSO enforces measurement-domain consistency, while the ISO sharpens image-domain details. We implement this by passing the output of each operator as a skip connection into the input of its counterpart, so that both branches evolve jointly:

$$y^{(n+1)} = \phi_\theta^y\Big(\alpha^{(n)} \cdot y^{(n)} + \beta^{(n)} \cdot \tilde{y}^{(n)}\Big), \qquad x^{(n+1)} = \phi_\theta^x\Big(\alpha^{(n)} \cdot x^{(n)} + \beta^{(n)} \cdot \tilde{x}^{(n)}\Big). \tag{16}$$

where $\phi_\theta^y$ and $\phi_\theta^x$ are refinement modules, and $\alpha^{(n)}$ and $\beta^{(n)}$ are learned scalar gates at scale $n$, with $n$ indexing the IFIB stage within the encoder–decoder hierarchy.

To boost performance, we adopt a refinement block (RB) that applies learned priors to stabilize and refine the transformed features, enabling coarse-to-fine reconstruction. Following Chen et al. (2022), the RB is a normalization-free residual module built from depthwise-separable convolutions with a simple channel-gating mechanism. Lightweight convolutional layers are placed before and after the core to stabilize the feature statistics produced by the forward–inverse integration. Further ablations on FSO/ISO design, their integration, and the RB are reported in Appendix A.5.

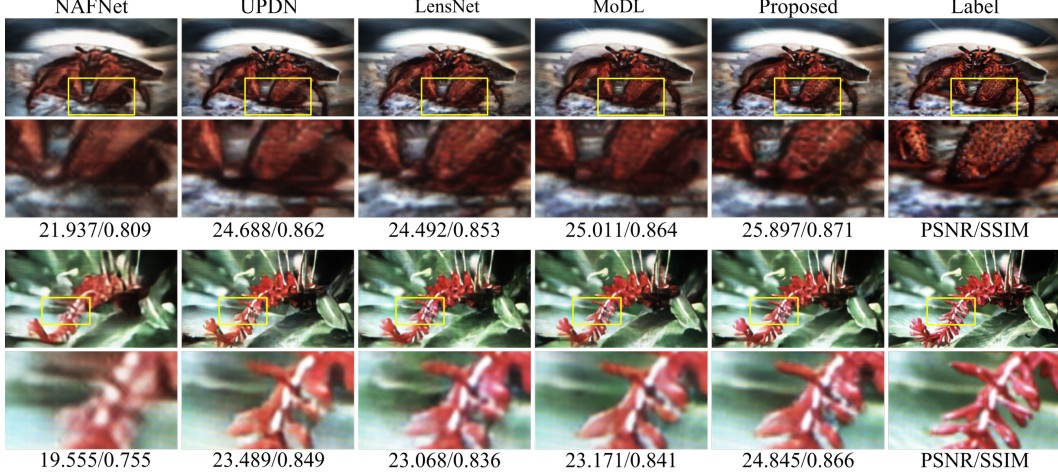

Figure 3: Visual comparison on DiffuserCam display-capture data. IFIN preserves color fidelity and high-frequency textures while suppressing artifacts. Insets mark zoomed regions and structures.

## 5 Results

To evaluate IFIN, we conduct end-to-end supervised training on large sets of scene–measurement pairs spanning both real display-capture and synthetic data, and compare against classical methods (Wiener deconvolution, ADMM-TV), a standalone ISO baseline, data-driven models (U-Net,

Table 1: Quantitative comparison on three benchmarks—**DiffuserCam**, **SV Lensless**, and **Multi-WienerNet**. We report PSNR ↑, LPIPS ↓ (Zhang et al., 2018a), and SSIM ↑ (arrows indicate the preferred direction). We compare classical methods, a standalone ISO baseline, data-driven models, hybrid approaches, and IFIN. Best in bold, second best underlined.

| Dataset | DiffuserCam | | | SV Lensless | | | MultiWienerNet | | |
|---|---|---|---|---|---|---|---|---|---|
| Metrics | PSNR ↑ | LPIPS ↓ | SSIM ↑ | PSNR ↑ | LPIPS ↓ | SSIM ↑ | PSNR ↑ | LPIPS ↓ | SSIM ↑ |
| ADMM | 12.252 | 0.607 | 0.346 | 11.843 | 0.643 | 0.323 | 19.189 | 0.557 | 0.420 |
| Wiener Deconv. | 12.552 | 0.591 | 0.384 | 12.405 | 0.607 | 0.369 | 18.658 | 0.640 | 0.302 |
| **ISO** | 16.528 | 0.544 | 0.404 | 17.240 | 0.462 | 0.444 | 20.202 | 0.623 | 0.380 |
| UNet | 21.230 | 0.394 | 0.656 | 21.890 | 0.474 | 0.646 | 23.859 | 0.389 | 0.589 |
| NAFNet | 24.830 | 0.239 | 0.810 | 23.857 | 0.245 | 0.769 | 24.657 | 0.282 | 0.712 |
| Le-ADMM-U | 23.261 | 0.312 | 0.765 | 21.956 | 0.278 | 0.748 | 23.732 | 0.335 | 0.702 |
| DeepLIR | 25.958 | 0.260 | 0.829 | 20.523 | 0.339 | 0.642 | 22.556 | 0.379 | 0.642 |
| MWNet | 24.832 | 0.247 | 0.810 | 23.001 | 0.255 | 0.766 | 25.660 | 0.260 | 0.728 |
| UPDN | 28.228 | 0.194 | 0.877 | 23.920 | 0.229 | 0.801 | 24.364 | 0.287 | 0.707 |
| MWDN | 27.298 | 0.217 | 0.845 | 24.525 | 0.224 | 0.801 | 27.436 | 0.236 | 0.780 |
| LensNet | 27.650 | 0.201 | 0.868 | 24.615 | 0.219 | 0.806 | 27.546 | 0.221 | 0.809 |
| MoDL | 27.958 | 0.183 | 0.878 | 24.791 | 0.202 | 0.810 | 28.504 | 0.202 | 0.831 |
| **IFIN (Proposed)** | **29.862** | **0.174** | **0.893** | **25.444** | **0.201** | **0.824** | **31.083** | **0.175** | **0.866** |

NAFNet), and hybrid approaches (Le-ADMM-U, DeepLIR, MWNet, UPDN, MWDN, LensNet, MoDL). Details of baseline implementations, evaluation protocols, and datasets are provided in Appendix A.2, Appendix A.3, and Appendix A.4.

**DiffuserCam** (Monakhova et al., 2019)—display-capture measurements acquired with a diffuser-based lensless camera, using co-located reference camera images as ground truth. As summarized in Table 1, IFIN achieves the best PSNR, LPIPS, and SSIM on this benchmark. Figure 3 shows that IFIN preserves color fidelity and fine textures, whereas competing methods tend to oversmooth details or introduce artifacts, especially near edges and in low-contrast regions.

**Custom Shift-Variant (SV) Lensless**—display-capture data acquired using our custom-designed lensless camera. Our in-house dataset has a wide field of view (FoV), exhibiting strong PSF shift variance in the outer regions but sharp, high-resolution structure near the center. IFIN successfully handles both regimes: it restores peripheral regions through explicit shift-variant modeling while faithfully preserving the sharp central details, without requiring direct measurement of off-axis PSFs. As shown in Figure 4, this leads to sharper, less distorted reconstructions across the full FoV compared to competing approaches, and Figure 5 further illustrates robustness on in-the-wild captures.

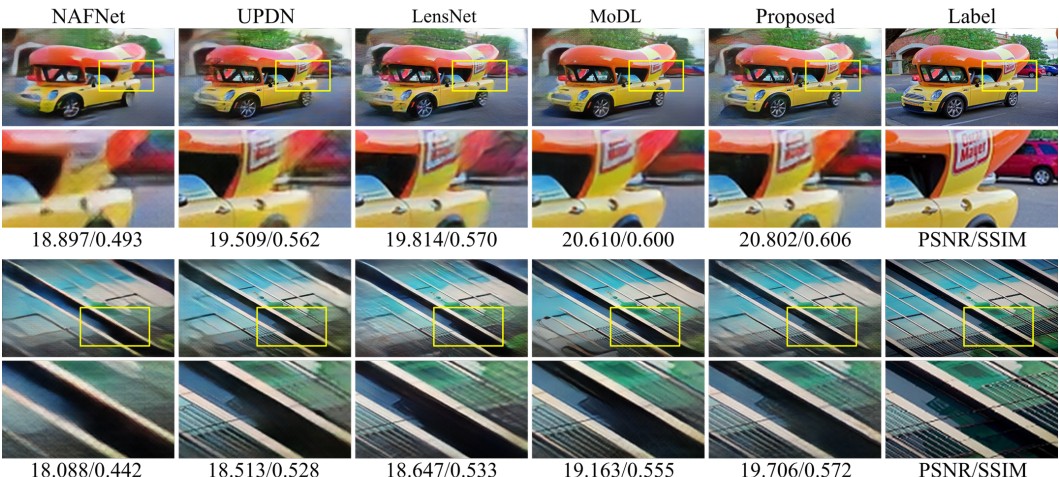

Figure 4: Comparison on SV lensless dataset. IFIN reduces peripheral blur and distortion while retaining fine details across the field.

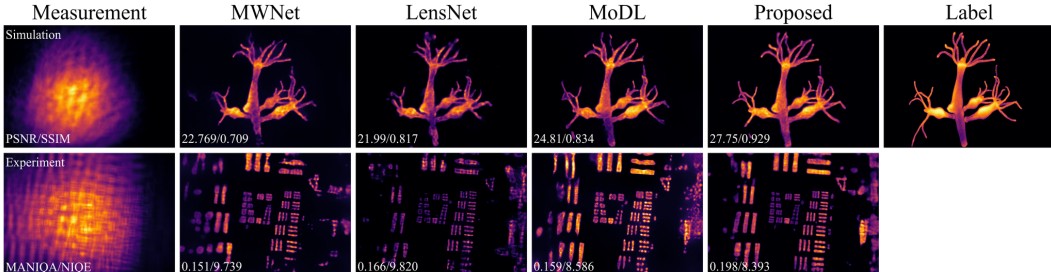

Figure 5: In-the-wild SV Lensless measurements without ground truth. IFIN generalizes to diverse scenes and lighting, reducing ringing and color shifts while preserving edges and textures. We report MANIQA↑ (Yang et al., 2022) and NIQE↓ (Mittal et al., 2012).

**MultiWienerNet (MW)** (Yanny et al., 2022)—synthetic training pairs are generated by convolving ground-truth labels with spatially variant PSFs measured from a mask-based microscope (miniscope), with validation performed on real captures. On the MW dataset, IFIN yields the sharpest and most faithful reconstructions in both simulated and real measurements (Figure 6). In simulated data, IFIN closely matches the ground-truth structure and contrast, achieving the best PSNR/SSIM, while on real miniscope measurements it preserves a wide effective FoV and sharp USAF targets with fewer ringing artifacts and better separation of closely spaced line pairs than competing methods. Notably, MWNet reconstructs with nine calibrated PSFs, which explains its advantage over approaches that rely on a single on-axis PSF. In contrast, IFIN is initialized with only a single on-axis PSF and learns effective off-axis PSFs during training (see Appendix A.6 for details), enabling it to handle shift variance without dense per-field calibration. As a result, whereas MoDL remains strong in simulation but degrades on experimental data, IFIN adapts more reliably to the real forward model.

Figure 6: Comparison on MW dataset. The first row shows simulated spatially variant measurements and reconstructions; the second row shows experimental miniscope captures of a USAF target.

## 6 CONCLUSION

We proposed **IFIN**, a physics-guided reconstruction network that integrates forward and inverse operators at every scale with an adaptively learnable PSF field. IFIN maintains two coupled streams throughout the network—a measurement stream constrained by the Forward System Operator (FSO) and an image stream refined by the Inverse System Operator (ISO)—where FSO and ISO implement FFT-based forward and inverse models derived from the optical PSF, so that measurement information is preserved and forward–inverse consistency is enforced in feature space. Experiments across DiffuserCam, SV Lensless, and MultiWienerNet benchmarks show that this symmetric forward–inverse integration, together with the multi-scale PSF field, enables high-fidelity recovery under large-support, shift-variant kernels, including settings without calibrated PSFs where kernels are learned end-to-end from data.

## 7 ETHICS STATEMENT

This study presents an advanced reconstruction method for inverse problems with potential relevance to domains such as medical imaging, security inspection, and optical system design. We emphasize responsible research practices by carefully considering ethical implications. Our experiments do not involve human subjects, personal data, or sensitive information; instead, the datasets are either synthetically generated or collected under controlled settings to avoid privacy risks. While the proposed method could in principle be applied to biomedical or security-related contexts, the present work is intended solely for scientific progress. Any practical deployment in sensitive areas must comply with applicable ethical standards and obtain proper regulatory approvals. Furthermore, we consider and address potential biases in both data and models to promote fairness, robustness, and broad generalizability across diverse imaging conditions. By following these principles, our research aims to contribute responsibly to computational imaging and the broader study of reconstruction.

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

# A APPENDIX

## A.1 TRAINING DETAILS

We train IFIN using the AdamW optimizer with a learning rate of $1 \times 10^{-4}$ and $(\beta_1, \beta_2) = (0.9, 0.999)$ for all parameters except the point spread function (PSF). For the PSF, we employ a separate AdamW optimizer with a learning rate of $1 \times 10^{-3}$. Both learning rates are reduced by a factor of $0.5$ when the validation loss plateaus. The training is conducted with a batch size of $4$.

To initialize the network, for spatially varying deconvolution we construct ROI maps aligned to the input size, enabling local adaptation without excessive computational overhead. In the forward–inverse integration, the weighting parameters are initialized as $\alpha = 0.8$ and $\beta = 0.2$, balancing the contributions of the forward and inverse operators at the start of training.

For the PSF representation, to reduce computational cost we crop the support region of the on-axis PSF to include only its effective feature area. Under shift-variant conditions, we do not rely on calibrated PSFs; instead, the on-axis PSF is either replicated $k$ times or randomly initialized. We experiment with $k = \{1, 4, 9, \mathbf{16}\}$ on DiffuserCam, and set $k = 9$ for both SV Lensless and MW datasets.

To ensure fairness, we match hyperparameters to those used in prior works. Whenever a network requires PSF inputs, we apply proper normalization (e.g., $\ell_2$, $\ell_1$, or max normalization) for stable training and meaningful results. Dataset-specific preprocessing steps such as cropping or affine

registration are not incorporated into the loss, but only applied for visualization. Further details on dataset preparation and affine transforms are provided in A.3 and A.4

Training IFIN on DiffuserCam requires 144 hours for $k = 16$ using two NVIDIA A6000 GPUs. As $k$ decreases, the training time scales down to 112 ($k = 9$), 92 ($k = 4$), and 74 hours ($k = 1$).

**Loss Functions.** We minimize a composite objective balancing pixel fidelity, perceptual quality, cross-domain consistency, and a physics prior on the PSF. Let $y$ be the observed input, $x$ the ground-truth, $\hat{x}$ the final reconstructed output, $\hat{y}$ an intermediate output indicated as measurement, $h$ the learned PSF, $\tilde{x}^{(0)}$ the first ISO-branch output, and $\mathcal{I}(\cdot, h)$ the ISO operator with PSF $h$.

The image-domain loss $\mathcal{L}_{\text{img}}$ enforces pixel-wise fidelity between the reconstruction $\hat{x}$ and the ground truth $x$. It anchors training to a numerically accurate solution and prevents the network from drifting under purely perceptual or physics-driven objectives.

The perceptual loss $\mathcal{L}_{\text{perc}}$ is implemented as the VGG-based LPIPS distance (Zhang et al., 2018a) between $\hat{x}$ and $x$. By comparing images in a deep feature space rather than pixel space, it encourages reconstructions that preserve semantic and textural similarity, helping to recover sharper, more visually plausible details than $\ell_2$ alone.

The ISO supervision term $\mathcal{L}_{ISO}$ directly constrains the first ISO-branch output $\tilde{x}^{(0)}$ to match the ground truth, ensuring that the inverse operator learns a meaningful, data-consistent deconvolution even in isolation. This stabilizes early training and makes the ISO usable as a standalone direct inverse.

The measurement-consistency loss $\mathcal{L}_{\text{cons}_{\text{im}}}$ penalizes discrepancies between the synthesized measurement $\hat{y}$ and the observed input $y$, enforcing data fidelity in the sensor domain. This term discourages hallucinated structures that cannot be explained by the forward model and improves robustness under model mismatch.

The cross-domain consistency loss $\mathcal{L}_{\text{cons}_{\text{ft}}}$ enforces agreement between the image reconstructed from the synthesized measurement, $\mathcal{I}(\hat{y}, h)$, and the main output $\hat{x}$. By tying together the forward and inverse branches, it promotes internal consistency between measurement- and image-domain representations and regularizes the joint optimization of FSO, ISO, and the PSF field.

The PSF loss $\mathcal{L}_{\text{psf}}$ enforces non-negativity of the learned kernels, consistent with the interpretation of the PSF as an intensity impulse response. This physical constraint regularizes blind kernel learning and helps stabilize the deconvolution, reducing noise amplification and ringing artifacts.

$$\mathcal{L}_{\text{img}} = \|\hat{x} - x\|_2^2 \qquad \text{fidelity loss} \qquad (17)$$

$$\mathcal{L}_{\text{perc}} = \text{LPIPS}_{\text{VGG}}(\hat{x}, x) \qquad \text{perceptual loss (Zhang et al., 2018a)} \qquad (18)$$

$$\mathcal{L}_{ISO} = \|\tilde{x}^{(0)} - x\|_2^2 \qquad \text{ISO supervision} \qquad (19)$$

$$\mathcal{L}_{\text{cons}_{\text{im}}} = \|\hat{y} - y\|_2^2 \qquad \text{measurement consistency} \qquad (20)$$

$$\mathcal{L}_{\text{cons}_{\text{ft}}} = \|\mathcal{I}(\hat{y}, h) - \hat{x}\|_2^2 \qquad \text{cross-domain alignment} \qquad (21)$$

$$\mathcal{L}_{\text{psf}} = \|\min(h, 0)\|_1 \qquad \text{PSF non-negativity} \qquad (22)$$

$$\mathcal{L} = \lambda_{\text{img}}\mathcal{L}_{\text{img}} + \lambda_{\text{perc}}\mathcal{L}_{\text{perc}} + \lambda_{ISO}\mathcal{L}_{ISO} + \lambda_{\text{cons}_{\text{im}}}\mathcal{L}_{\text{cons}_{\text{im}}} + \lambda_{\text{cons}_{\text{ft}}}\mathcal{L}_{\text{cons}_{\text{ft}}} + \lambda_{\text{psf}}\mathcal{L}_{\text{psf}} \qquad (23)$$

$$(\lambda_{\text{img}}, \lambda_{\text{perc}}, \lambda_{ISO}, \lambda_{\text{cons}_{\text{im}}}, \lambda_{\text{cons}_{\text{ft}}}, \lambda_{\text{psf}}) = (1.0, 0.05, 0.1, 0.01, 0.01, 0.1). \qquad (24)$$

Coefficients are selected by validation and kept fixed across all experiments. For fairness, all baseline models that we retrain in our framework are optimized only with the image and perceptual losses ($\mathcal{L}_{\text{img}}$ and $\mathcal{L}_{\text{perc}}$), using the same loss weights for all datasets.

## A.2 BASELINES

We summarize each baseline used in our study, together with its original reference and role in our experiments.

**Wiener Deconvolution** (Wiener, 1964): Classical closed-form deconvolution using a calibrated PSF. We tune the noise-to-signal parameter on a validation split. It is fast and simple but sensitive to noise and kernel mis-specification.

**Alternating Direction Method of Multipliers (ADMM–TV)** (Boyd et al., 2011): Variational reconstruction with a total-variation prior solved via ADMM, which decouples data fidelity and regularization; we use shared stopping criteria across scenes.

**U-Net** (Ronneberger et al., 2015): A simple encoder–decoder CNN with skip connections that serves as a purely data-driven reconstruction baseline.

**NAFNet** (Chen et al., 2022): A modern, parameter-efficient CNN restorer built from Nonlinear Activation Free (NAF) blocks, which replace explicit nonlinear activations with lightweight gating and normalization. As a recent CNN architecture for image restoration, NAFNet provides a strong purely data-driven baseline and helps illustrate how a modern denoiser design compares with physics-driven or hybrid methods.

**Learned-ADMM-U (Le-ADMM-U)** (Monakhova et al., 2019): Unrolled ADMM with learnable proximal operators over $K$ iterations and U-Net denoiser. Its simple structure proposes a combination of physics and neural networks.

**DeepLIR** (Poudel & Nakarmi, 2024): Learned iterative reconstruction that updates with a learned denoiser over $K$ steps; the study utilizes the ConvNeXt blocks (Woo et al., 2023) in the denoiser with strong attention-based approach in convolutional layers.

**Unrolled Primal–Dual Network (UPDN)** (Kingshott et al., 2022): A primal–dual unrolling architecture in which the measurement and image variables are updated through repeated primal–dual layers containing CNN blocks, followed by a final encoder–decoder denoiser applied to the reconstructed image.

**MultiWienerNet (MWNet)** (Yanny et al., 2022): A lightweight hybrid baseline that linearly combines multiple Wiener-filter outputs and refines them with U-Net under spatially varying conditions. When calibration is unavailable, we use an on-axis PSF and instantiate $k$ filters accordingly for this baseline.

**Modular Learned Reconstruction (MoDL)** (Bezzam et al., 2025): A modular reconstruction framework that performs a learned-ADMM inverse step wrapped by encoder–decoder denoisers applied before and after the inverse; the PSF is also processed by a dedicated denoiser, improving robustness to noisy measurements and calibration.

**MultiWiener Deconvolution Network (MWDN)** (Li et al., 2023): Feature-space deconvolution within a multi-scale encoder–decoder utilizing PSF encoder: Wiener-like inversions are inserted along skip connections to improve fidelity and robustness under severe blur.

**LensNet** (Bai et al., 2025): A feature-space deconvolution baseline built on a multi-scale encoder–decoder with advanced CNN blocks, using randomly parameterized PSFs to perform blind deconvolution.

### A.3 DETAILED DESCRIPTION OF PUBLIC DATASETS

**DiffuserCam.** The DiffuserCam dataset (Monakhova et al., 2019) comprises 25,000 paired captures acquired simultaneously with a mask-based lensless camera (Antipa et al., 2017) and a reference lensed camera aligned via a beam splitter, using images from MIRFlickr (Huiskes & Lew, 2008) displayed on a computer monitor. The prototype uses an off-the-shelf Light Shaping Diffuser (Lu-

minit $0.5°$) with a laser-cut paper aperture, positioned approximately $9\,\text{mm}$ in front of the sensor plane. Raw sensor frames of $1080 \times 1920$ pixels are downsampled by a factor of four to $270 \times 480$. For visualization, DiffuserCam images are additionally cropped to $380 \times 210$, while all reported metrics (PSNR, SSIM, LPIPS) are computed on the non-cropped scenes. The dataset is split into 24,000 training and 1,000 test images. A single PSF is calibrated at the field center using an on-axis LED point source at the screen plane.

**MultiWienerNet.** Built on microscope data from Miniscope3D (Yanny et al., 2020), the Multi-WienerNet dataset (Yanny et al., 2022) explicitly calibrates spatially varying PSFs across the field: a sub-resolution bead is scanned to measure the PSF at multiple sensor locations, effectively sampling a $3 \times 3$ grid over the imaging field. Using these measured PSFs, a synthetic training set is generated by convolving natural images with the spatially varying forward model and adding Poisson and Gaussian noise to emulate realistic measurements. This yields 22,125 two-dimensional paired samples, split 80/20 for training and testing. All training data are simulated at the system's sensor field-of-view resolution, and the trained models are finally evaluated on real lensless measurements from the calibrated setup. In our experiments, we resize all images to $320 \times 224$ to reduce computational cost.

### A.4 SV Lensless Dataset

We introduce a new dataset built around a multi-lens-array-like phase mask engineered for a compact, high-resolution lensless camera. The mask profile is optimized for our optical design and fabricated via grayscale lithography. Using a Sony IMX708 sensor, we assemble the camera by placing a $2\,\text{mm}$ aperture and the phase mask at approximately $1.6\,\text{mm}$ from the sensor. By design, we prioritize on-axis resolution at the cost of increased PSF shift variance with incident angle. MIR-Flickr images are displayed such that they occupy approximately 80% of a 48-inch OLED TV screen at a working distance of $30\text{cm}$, as shown in Figure 7. A total of 25,000 images are captured and then split into 24,000 for training and 1,000 for testing, following the same indices of the DiffuserCam dataset. We capture $4608 \times 2592$ measurements from sensor and resize them to $480 \times 270$. For target alignment, we first reconstruct an image using a deconvolution baseline, estimate an affine transform between the reconstruction and its corresponding label, and apply this transform to the label images during training, as illustrated in Figure 8. Unlike the DiffuserCam dataset, where a separate reference camera and spatial cropping are used to align, compare, and visualize the labels, our setup performs alignment purely through the estimated affine transform applied to the label. The metrics including PSNR, SSIM, and LPIPS are measured using the affine-transformed label, and the inverse transform is applied only when visualizing the output.

**Optical Design Constraints** We designed the phase mask in a deductive manner under the mechanical and optical constraints. The constraints included: (i) RPI3 sensor module's mechanical stack and housing, (ii) target field of view (FoV) and equivalent focal length, and (iii) the maximum fabricable optical thickness. Within these constraints, we derived a single planoconvex micro-lens profile with 20 $\mu$m of vertex height and 860 $\mu$m radius of curvature that maximizes the effective numerical aperture (NA).

The unit profile was randomly tiled over a 3.5 mm $\times$ 3.5 mm area to form the phase mask. A minimum inter-lens spacing $d_{\min}$ was enforced during placement to preserve fill factor and suppress degradation of the effective NA due to mutual overlap and edge clipping.

**Fabrication.** The mask was fabricated as a multi-level phase element via grayscale lithography (Anonymous, 2023). The continuous height map of the planoconvex profile was converted to grayscale dose, enabling a single exposure-development process. Post-fabrication inspection (surface profiler/microscopy) verified profile integrity and absence of large-area defects.

**Camera Assembly.** The fabricated mask was aligned and attached in front of the RPI3 image sensor to form a lensless camera. A mechanical aperture of 2 mm diameter was applied directly at the mask plane to define the active pupil and to mitigate stray light and edge effects during imaging.

**PSF Measurement and Optical Performance** We characterized the system by measuring the PSFs on- and off-axis. A comparison between the designed mask, simulated PSFs, and experimentally captured PSFs is shown in Figure 9, highlighting the agreement between design and physical performance.

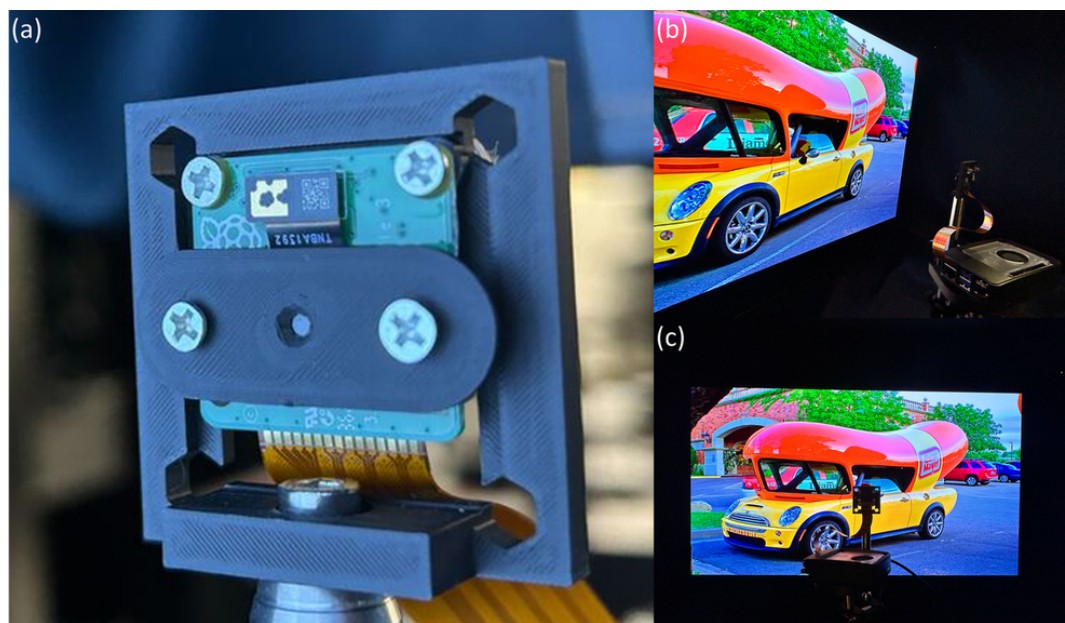

Figure 7: Prototype lensless camera and dataset capture setup. (a) Lensless camera prototype with a CMOS sensor mounted in a custom 3D-printed holder. (b–c) Display-capture configuration used for the SV Lensless dataset: reference images are rendered on a calibrated display while the prototype records the corresponding lensless measurements at a fixed geometry.

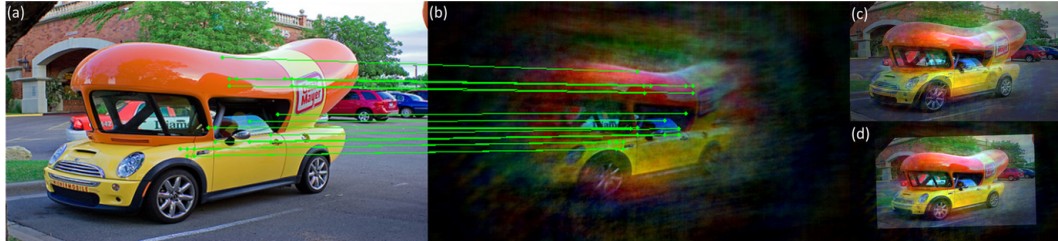

Figure 8: Affine registration for label and display-capture pairs. (a) Reference image shown on the display. (b) Raw lensless measurement captured by the prototype; green segments indicate LoFTR-based feature correspondences used to estimate the affine transform (Sun et al., 2021). (c) Overlay before registration. (d) Overlay after applying the estimated affine warp, yielding pixel-wise alignment suitable for supervised training and evaluation.

- **On-axis.** The measured PSF closely matched the designed PSF. Cross-correlation analysis with the design yielded high similarity, and the measured full width at half maximum (FWHM) was judged to be acceptable, consistent with the high effective NA. These results provide an empirical bound on the on-axis optical resolution.

- **Off-axis.** At larger field angles, aberrations became pronounced as expected under high-NA operation. The PSF exhibited asymmetric tails consistent with the coma aberration from the planoconvex element. Consequently, the system exhibits shift-variant imaging behavior: high optical performance on-axis, with aberration-limited quality off-axis.

**Implications** Maximizing NA under a limited thickness budget (20 $\mu$m) was effective for on-axis resolution but increases sensitivity to off-axis aberrations and FoV non-uniformity. Hardware routes to mitigate this include aspheric refinements, multi-layer phase designs, and orientation-aware cell geometries; software routes include deconvolution with a field-dependent PSF or learned reconstructions that explicitly model shift variance. As illustrated in Figure 10, deconvolving with PSFs

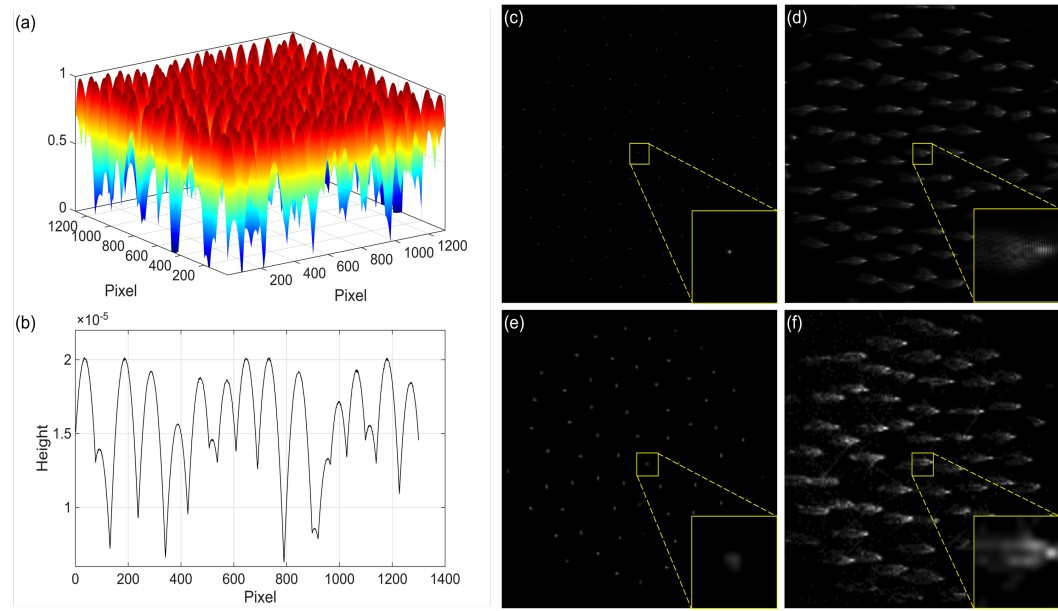

Figure 9: (a) The designed phase mask pattern optimized for high-resolution lensless imaging. (b) A representative line profile. (c) Simulated on-axis PSF (d) simulated off-axis PSF at $40°$, showing the effect of shift variance across the field. (e) captured on-axis PSF (f) captured off-axis PSF showing the effect of shift variance across the field.

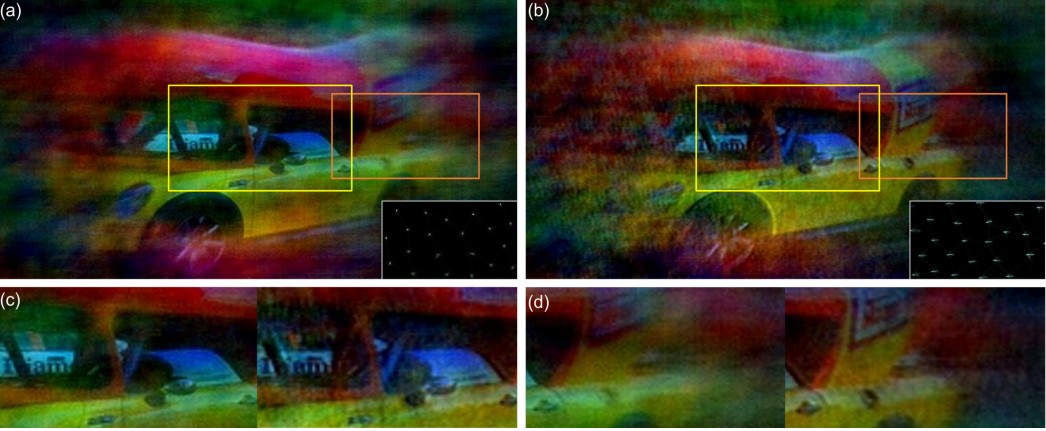

Figure 10: (a) Reconstruction using the center PSF along with the corresponding center PSF. (b) Reconstruction using an off-axis PSF along with the corresponding off-axis PSF. (c) Comparison of sharpness at the image center between (a) and (b). (d) Comparison at the image periphery, demonstrating differences caused by field-dependent PSFs.

drawn from different field regions produces noticeable differences in focal sharpness (and associated artifacts), directly evidencing the field dependence.

## A.5 ABLATION

**Effect of FSO/ISO and their integration.** We assess the contribution of explicitly modeling the forward and inverse processes (FSO/ISO) and the role of bidirectional exchange between them (Table 2).

*FSO/ISO as identity.* We replace both FSO and ISO with identity mappings and allow only latent mixing between the two streams. This control demonstrates that simple feature mixing is not sufficient: improvements are limited compared to the full IFIB, highlighting that enforcing forward and inverse operators within the feature space is crucial for propagating measurement-domain cues and stabilizing high-frequency reconstruction.

*w/o ISO.* We remove the inverse operators and retain only a forward-guided pathway. Without ISO, information flows primarily from the measurement domain, reducing feature sharpening and attenuating high-frequency components. This results in softer textures and lower PSNR/SSIM.

*w/o FSO.* We remove the forward operators and retain only inverse guidance, where the measurement is injected only at the first stage of the network. Without FSO to enforce explicit physics in the feature space, the model shows a noticeable drop in fidelity and less stable convergence, although it still benefits from the learned inverse pathway.

Table 2: Ablation on forward/inverse modeling and their integration on DiffuserCam (PSNR $\uparrow$ / LPIPS $\downarrow$ / SSIM $\uparrow$). We compare: (i) Identity FSO/ISO (no physics, feature mixing only), (ii) w/o ISO (forward-only guidance), (iii) w/o FSO (inverse-only guidance), and (iv) Proposed (full IFIB with forward–inverse integration).

| Setting ($k{=}16$) | PSNR $\uparrow$ | LPIPS $\downarrow$ | SSIM $\uparrow$ |
|---|---|---|---|
| ISO / FSO as identity | 24.674 | 0.255 | 0.800 |
| w/o ISO | 27.123 | 0.223 | 0.833 |
| w/o FSO | 28.711 | 0.185 | 0.882 |
| **Proposed** | **29.862** | **0.174** | **0.893** |

**Effect of internal components in FSO and ISO.** We next ablate internal design choices in FSO and ISO, including the learnable 2-D regularization term and padding strategies used to mitigate boundary artifacts (Table 3). Removing the 2-D regularizer or omitting padding in either operator consistently degrades PSNR/SSIM and increases LPIPS, confirming that stable frequency-domain inversion and carefully handled boundaries are both important for high-fidelity reconstruction.

Table 3: Ablation of internal components in ISO and FSO (DiffuserCam, $k{=}1$). We evaluate the impact of removing the 2-D regularization term and padded operation at FSO (zero-padding) and ISO (replicate-padding).

| Setting ($k{=}1$) | PSNR $\uparrow$ | LPIPS $\downarrow$ | SSIM $\uparrow$ |
|---|---|---|---|
| w/o 2-D regularizer | 28.880 | 0.186 | 0.881 |
| w/o padding in ISO & FSO | 28.445 | 0.191 | 0.879 |
| w/o padding in ISO | 29.001 | 0.185 | 0.883 |
| w/o padding in FSO | 29.037 | 0.184 | 0.885 |
| **Proposed** | 29.535 | 0.179 | 0.892 |

**Effect of the RB.** We compare the proposed refinement block (RB) against a simpler convolutional baseline (ConvG), which uses a shallow convolution–gating structure without the normalization-free residual design of RB. As shown in Table 4, replacing RB with basic, and simpler ConvG used in PSF encoder leads to a noticeable drop in PSNR/SSIM and higher LPIPS, indicating that the stronger prior encoded by RB is beneficial for stabilizing the forward–inverse integration and refining high-frequency details.

**Effect of the PSF encoder.** We also ablate the PSF encoder by replacing it with simple resizing of the calibrated PSF without any CNN processing (Table 5). The CNN-based encoder yields consistently higher PSNR/SSIM and lower LPIPS, demonstrating that learning multi-scale PSF embeddings from data is preferable to using a fixed, hand-crafted representation.

**Effect of the number of PSFs.** We vary the number of learnable PSFs $k = s^2$. Small $k$ (e.g., $s{=}1$) underfits spatial variability, while excessively large $k$ increases computation without propor-

Table 4: Ablation of the refinement block (RB) on DiffuserCam. ConvG denotes a lightweight baseline block composed of a Conv2D layer, LayerNorm, and GELU activation, and RB is the block with normalization-free residual module utilized in proposed model.

| Block configuration | PSNR ↑ | LPIPS ↓ | SSIM ↑ |
|---|---|---|---|
| ConvG | 29.014 | 0.184 | 0.882 |
| **RB (Proposed)** | **29.535** | **0.179** | **0.892** |

Table 5: Ablation of PSF modeling in the proposed method with $k{=}1$ (DiffuserCam). We compare simple resizing without a CNN encoder and the full PSF encoder.

| PSF configuration | PSNR ↑ | LPIPS ↓ | SSIM ↑ |
|---|---|---|---|
| resize only | 29.014 | 0.182 | 0.886 |
| **CNN encoder (Proposed)** | **29.535** | **0.179** | **0.892** |

tional gains. In particular, because the computational cost of the ISO branch scales linearly with $k$, selecting $k$ becomes especially important when the PSF kernels have large spatial support.(Table 6).

Table 6: Ablation on the number of learnable PSFs ($k = s^2$) across the three benchmarks.

| Dataset | DiffuserCam | | | SV Lensless | | | MultiWienerNet | | |
|---|---|---|---|---|---|---|---|---|---|
| Metrics | PSNR ↑ | LPIPS ↓ | SSIM ↑ | PSNR ↑ | LPIPS ↓ | SSIM ↑ | PSNR ↑ | LPIPS ↓ | SSIM ↑ |
| $k{=}1$ | 29.535 | 0.179 | 0.892 | 24.963 | 0.211 | 0.815 | 29.733 | 0.197 | 0.848 |
| $k{=}4$ | 29.612 | 0.177 | 0.893 | 25.148 | 0.205 | 0.820 | 30.208 | 0.185 | 0.855 |
| $k{=}9$ | 29.751 | 0.176 | 0.893 | 25.444 | 0.201 | 0.824 | 31.083 | 0.175 | 0.866 |

**Effect of the initial PSF.** We compare initializing the PSF with a calibrated on-axis measurement versus random noise (Table **??**). Using the calibrated PSF generally leads to faster and more stable convergence, and the performance gap in terms of PSNR/SSIM remains modest. Importantly, however, calibration is not strictly required: even with random initialization, as long as the forward system is known and training pairs are available, the network can learn effective kernels in a fully data-driven blind deconvolution setting.

**Effect of the ROI scale $\sigma$.** We study the spatial scale of the Gaussian ROI maps used to blend region-wise PSFs. Table 7 reports a quantitative ablation where, for each choice of $\sigma$, we train a separate model from scratch. Too small $\sigma$ makes the ROIs overly local and fragmented, whereas too large $\sigma$ spreads each ROI too broadly, reducing the effective dynamic range of the blending weights and blurring the result. A moderate value (e.g., $\sigma_x{=}60$ in our setting for DiffuserCam) yields the best trade-off between fidelity and stability.

To better visualize the effect of $\sigma$ itself, Figure 11 shows a complementary experiment in which we freeze a trained ISO (and all other network weights) and vary only $\sigma$ at test time when generating the Gaussian ROIs. The qualitative trends are consistent with the retrained models: small $\sigma$ produces block-like artifacts and seams at tile boundaries, very large $\sigma$ compresses the dynamic range of the ROI weights and slightly blurs details, while the default setting and the learned ROI maps produce sharper and more spatially consistent reconstructions.

## A.6 LEARNED PSFs WITHOUT CALIBRATION

Figure 12 visualizes the learned 3×3 PSF field from MW dataset. Near the optical center, kernels are compact and approximately isotropic, whereas off-axis locations exhibit increased spread, slight centroid shifts, and mild anisotropy—patterns commonly observed with diffusers and wide-aperture optics. This spatial trend correlates with the improvements seen on real data: FSO reproduces location-dependent blurs using the learned PSFs, and ISO inverts them with data-driven regularization, leading to sharper reconstructions with fewer boundary artifacts.

Table 7: Ablation on the ROI Gaussian scale $\sigma_x$ (DiffuserCam, $k{=}16$).

| $\sigma_x$ in ROI ($k{=}16$) | PSNR ↑ | LPIPS ↓ | SSIM ↑ |
|---|---|---|---|
| 30 | 29.674 | 0.177 | 0.892 |
| 60 (Proposed) | 29.862 | 0.174 | 0.893 |
| 120 | 29.729 | 0.177 | 0.893 |

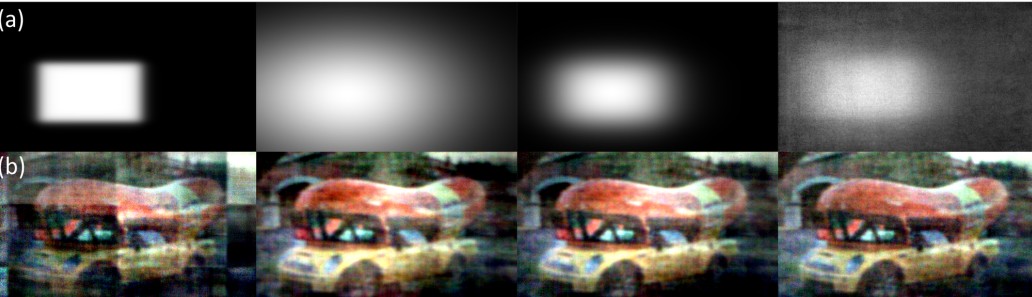

Figure 11: Effect of the ROI Gaussian scale on PSFs and reconstructions (DiffuserCam, $k{=}16$). (a) Examples of ROI maps for different settings: small $\sigma$, large $\sigma$, the default fixed $\sigma{=}60$ along the $x$-axis, and the learned ROI maps. (b) Corresponding reconstructions using each ROI configuration (columns aligned with (a)). In all cases, we use the same trained ISO and only the ROI scale $\sigma$ is modified at test time. Small $\sigma$ leads to fragmented regions and visible seams, large $\sigma$ oversmooths details, whereas the default and learned ROIs provide sharper and more spatially consistent results.

Notably, the learned PSFs remain normalized and vary smoothly across neighbors, reflecting physically plausible optics. Because the PSF field is shared across scales and injected into every IFIB, the network preserves forward–inverse consistency throughout the hierarchy. Qualitatively, these PSFs agree with expected diffuser patterns and reveal off-axis blur variations that standard shift-invariant models fail to capture, explaining IFIN's robustness under strong shift variance.

## A.7 COMPARISON OF DECONVOLUTION AND ISO

Figure 13 qualitatively compares classical Wiener deconvolution with our learned Inverse System Operator (ISO). On DiffuserCam, even when the deconvolution hyperparameters are carefully tuned, classical Wiener filtering only partially restores fine structure and tends to leave residual blur and ringing, whereas ISO produces sharper textures and cleaner edges under the same forward model. For the SV Lensless dataset, deconvolution is further limited by strong shift variance from the wide-FoV design and angular-response effects that cause peripheral light loss; in this regime, ISO better compensates for these spatially varying degradations and yields more uniform reconstructions across the field. On the MW dataset, ISO recovers a wider range of simulated structures and resolves tighter USAF target patterns than deconvolution, illustrating the benefit of incorporating system-aware operators into the network.

Beyond serving as a module inside IFIN, the proposed ISO can also be used as a standalone pre-trained inverse mapping, providing a fast, learned alternative to hand-tuned deconvolution for direct inference.

## A.8 GAUSSIAN DEBLUR SIMULATION.

To assess robustness beyond lensless imaging that uses specific large-support kernels, we synthesize a dataset with strong, lens-based optical blur. Clean natural images are convolved with Gaussian PSFs with $\sigma \in \{5, 10, 15\}$ to emulate heavy defocus. The dataset comprises 24,000 training images and 1,000 test images at $256 \times 256$ resolution, isolating the network's ability to handle large kernel support in a conventional deblurring task. See Table 8 for quantitative results. Compared with two strong baselines—RCAN (Zhang et al., 2018b) and NAFNet (Chen et al., 2022)—our

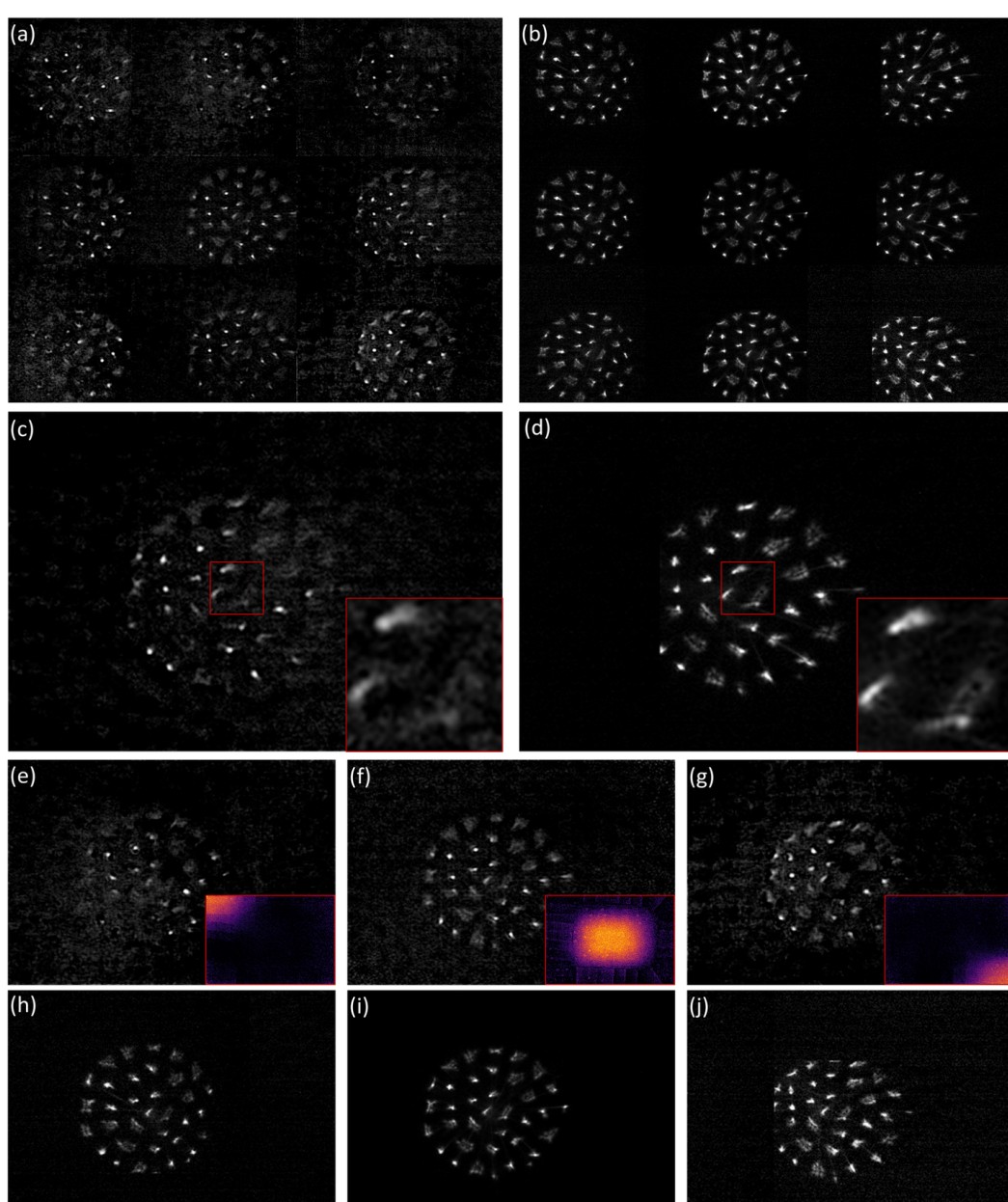

Figure 12: Comparison of learned and calibrated PSFs in MultiWienerNet Dataset (a) PSFs estimated through training. (b) Calibrated PSFs. (c) Estimated PSF at $r = 6$ near the right-center position. (d) Calibrated PSF at the corresponding location. (e–g) Estimated PSFs at different positions according to the learned ROI weights. (h–j) Calibrated PSFs corresponding to the same indices as in (e–g).

method achieves competitive performance at mild blur and becomes increasingly advantageous as blur severity grows. In our lensless experiments, the forward models are typically more structured and system-aware, so we expect the benefits of tightly integrating forward–inverse operators to be at least as pronounced as in this synthetic deblurring setting.

**Computational cost.** Spatially varying operators improve fidelity but reduce parameter sharing, which naturally increases memory usage and runtime relative to the shift-invariant case. To quantify this, we benchmark all methods with batch size 1 on a single NVIDIA RTX A6000 GPU, measuring FLOPs, peak GPU memory, and wall-clock inference time for a single forward pass. The results are

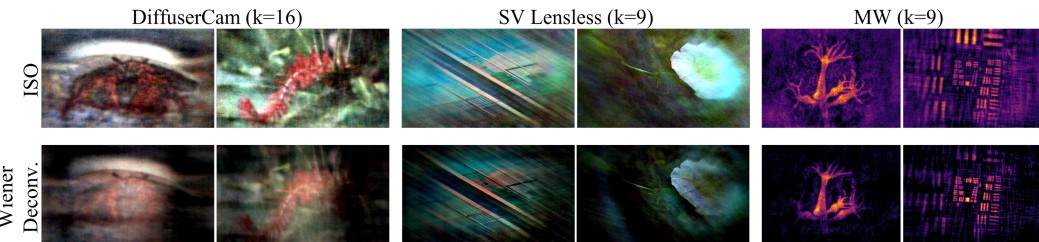

Figure 13: Comparison between the proposed ISO and classical Wiener deconvolution on Diffuser-Cam, SV Lensless, and MW. ISO mitigates noise amplification and ringing while improving fine-detail recovery.

Table 8: Quantitative comparison (PSNR ↑ / SSIM ↑ / LPIPS ↓) on synthetic Gaussian blur with noise levels $\sigma \in \{5, 10, 15\}$. Baselines (RCAN, NAFNet) vs. proposed method. Higher PSNR/SSIM and lower LPIPS indicate better restoration under increasing blur/noise.

| Method | $\sigma = 5$ | | | $\sigma = 10$ | | | $\sigma = 15$ | | |
|---|---|---|---|---|---|---|---|---|---|
| | PSNR ↑ | LPIPS ↓ | SSIM ↑ | PSNR ↑ | LPIPS ↓ | SSIM ↑ | PSNR ↑ | LPIPS ↓ | SSIM ↑ |
| RCAN | 25.303 | 0.220 | 0.810 | 22.623 | 0.330 | 0.750 | 21.280 | 0.355 | 0.711 |
| NAFNet | **25.625** | **0.205** | **0.825** | **22.810** | **0.315** | **0.766** | 21.535 | 0.345 | 0.715 |
| **Proposed** | 25.100 | 0.230 | 0.800 | 22.732 | 0.337 | 0.741 | **21.654** | **0.340** | **0.721** |

summarized in Table 9. While increasing the number of learnable PSFs $k$ raises the complexity of IFIN, moderate values remain comparable in FLOPs to other hybrid baselines and provide the best trade-off between quality and cost (see Table 6 for the corresponding accuracy trends).

The higher memory footprint and slower runtime of IFIN relative to baselines stem mainly from the FFT-based convolutions and deconvolutions used in FSO and ISO. These operators require padding to avoid wrap-around artifacts, and both the padded spatial support and the need to maintain multiple PSF-specific reconstructions in the ISO branch cause the cost to scale with $k$ and kernel size. To mitigate this, we crop each PSF to a tight region of interest around its effective support. This design keeps the overhead manageable while retaining most of the gains from spatially varying modeling.

IFIN consistently outperforms other baselines in reconstruction quality across all tested settings, including the shift-invariant case ($k$=1), so its computational cost should be interpreted in light of this accuracy gain. We view further optimization of FFT-based operators and convolution strategies as an important direction for reducing the runtime and memory overhead of such hybrid models.

Table 9: Complexity comparison across architectures on DiffuserCam. We report FLOPs, peak memory, and inference time per image (batch size 1) on an NVIDIA RTX A6000 GPU.

| Method | FLOPs (G) | Peak Memory (GB) | Inference Time (ms) |
|---|---|---|---|
| NAFNet | 32.4 | 0.257 | 24.65 |
| MWDN | 106.4 | 0.422 | 12.81 |
| UPDN | 24.5 | 0.326 | 60.66 |
| LensNet | 147.2 | 0.529 | 43.00 |
| MoDL | 119.1 | 0.202 | 36.60 |
| IFIN ($k$=1) | 55.0 | 0.464 | 142.35 |
| IFIN ($k$=4) | 62.4 | 1.257 | 156.33 |
| IFIN ($k$=9) | 82.4 | 2.614 | 180.37 |
| IFIN ($k$=16) | 126.9 | 4.590 | 215.97 |

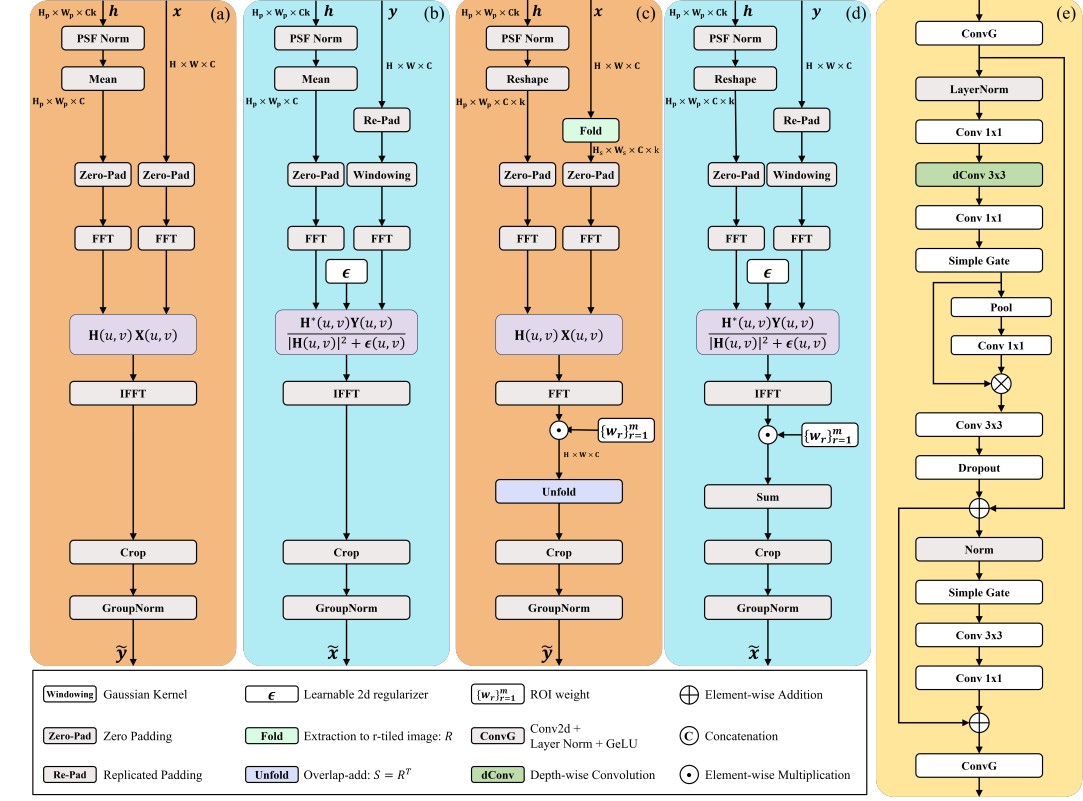

Figure 14: Detailed flow of system operators and the refinement block used in IFIB. (a) Basic FSO. (b) Basic ISO. (c) Fully-SV FSO. (d) SV ISO. (e) Refinement block.

### A.9 MODELING SHIFT VARIANCE IN FSO

We also considered a fully shift-variant formulation of the forward operator, analogous to ISO, by decomposing $\hat{x}$ into overlapping tiles, padding each tile to the local PSF support, convolving locally, and reassembling via normalized overlap-add:

$$\tilde{y} = \sum_{r=1}^{m} S_r\Big(\big(R_r\hat{x}\big) * h_r\Big), \qquad (25)$$

where $R_r$ extracts the $r$-th tile and $S_r = R_r^{\top}$ denotes overlap-add with normalization by the local coverage count to avoid seams. In practice, however, this design significantly increases computational overhead despite efforts to optimize tiling. Moreover, the forward operator in IFIN primarily serves to preserve measurement-domain properties rather than to synthesize high-fidelity outputs. Providing it in a simplified, shift-invariant (averaged) form reduces model mismatch while retaining the necessary measurement consistency signal. For these reasons, we adopt the shift-invariant forward operator in our main design. We anticipate that more precise yet simplified variants of the forward operator can be integrated when available, further improving fidelity without incurring significant overhead. A detailed flow of the system operators, including the shift-variant FSO, is provided in Figure 14.

### A.10 EXPANSION OF SYSTEM VARIANCE

Depending on the optical configuration, the same PSF field can be re-indexed along the axis.

Concretely, we write

$$h_r[\Delta i, \Delta j; q], \quad r = 1, \ldots, s^2, \; q \in \mathcal{Q},$$

where $r$ indexes lateral regions (field dependence) and $q$ indexes the nonlateral axis (e.g., depth $z$, wavelength $\lambda$, or time $t$). For scenarios where the PSF varies both laterally and axially, we adopt a 5-D depth-space-variant PSF:

$$h_{i,j}[\Delta k, \Delta \ell; \, z],$$

The corresponding forward model is

$$y[i,j] \; = \; \sum_{z=1}^{N_z} \sum_{k,\ell} h_{i,j}[k, \ell; \, z] \; x[k, \ell, z] \; + \; \eta[i,j], \qquad (26)$$

where $x[\cdot, \cdot, z]$ denotes the scene slice at channel $z$. For purely field-dependent blur we set $|\mathcal{Q}|=1$ and recover the 2-D case. This re-indexing keeps the forward/inverse operators unchanged in form while allowing IFIN to adapt the PSF dimension to the underlying system; in this work we focus on the 2-D shift-variant setting.

### A.11 DISCUSSION AND LIMITATION

Across the three benchmarks (DiffuserCam, SV Lensless, and MultiWienerNet), IFIN achieves the best scores in all metrics (Table 1). In terms of PSNR, it improves over the strongest prior learning-based method on each dataset (UPDN on DiffuserCam, MoDL on SV Lensless and MultiWienerNet) by +1.63 dB, +0.65 dB, and +2.58 dB, respectively. These gains are accompanied by consistently lower LPIPS and higher SSIM, indicating that the improvements reflect both sharper details and better perceptual quality. The advantages are particularly pronounced near the field periphery and on the MW benchmark, where large PSFs severely degrade purely CNN-based inversion.

We attribute these improvements to the way the reconstruction network is built around the known forward model and its inverse. Rather than relying on a single inversion stage, IFIN utilizes the integrated forward–inverse with learned feature transforms so that measurement-domain consistency is enforced throughout the feature hierarchy, while learnable shift-variant operators driven by a multi-scale PSF field allow the network to adapt to system mismatch. Taken together, these components define a flexible architectural template for constructing reconstruction networks directly from a given forward–inverse pair, which can be flexibly instantiated for convolution/deconvolution-based lensless imaging systems and other modalities that admit similar models.

**Generality beyond a single modality.** The integrated forward–inverse formulation is not tied to a particular optical prototype. Any system that can be approximated by a convolutional model— for example, defocus or motion blur in photography, field-dependent aberrations in microscopy or telescopy, or atmospheric turbulence in astronomy—could be handled by replacing the PSF field and operators in IFIN with the corresponding system model. In that sense, the proposed blocks provide a general recipe for embedding forward and inverse physics into feature space without redesigning the overall architecture for each modality.

This apparent generality, however, rests on several modeling assumptions. First, we assume that the forward operator can be well approximated by a discretized convolutional (or pseudo-convolutional) model with a calibrated PSF field, and that the sampling pattern and noise statistics used in training reasonably match those encountered at inference. In the present implementation, measurement noise is handled implicitly via regularizer in the inverse step. Second, the formulation requires access to both a forward operator and a corresponding pseudo-inverse that are sufficiently accurate and differentiable to be embedded as layers. When these assumptions are significantly violated—for instance, in the presence of strong nonlinear sensor effects, severe model mismatch, missing or inaccurate knowledge of the forward physics, or noise regimes that deviate substantially from those implicitly encoded in the regularizer—the reconstruction quality may degrade and the learned behavior may deviate from the intended physics. In such cases, IFIN should be regarded as a flexible template rather than a fixed design: the same block structure can accommodate alternative forward operators, approximate inverses, or more explicit noise-aware components that better reflect the target modality, without requiring a complete redesign of the surrounding network.

**Potential extensions.** Several extensions follow naturally. One direction is to move beyond a single 2D PSF field and incorporate depth-, wavelength-, or time-dependent operator channels, so that volumetric, hyperspectral, or dynamic imaging can be handled within the same IFIN template. In this setting, the PSF field becomes a multi-channel quantity indexed not only by spatial position

but also by depth, color, or temporal coordinates, and the corresponding forward–inverse operators act jointly across these channels. Another direction is to impose low-rank or separable structure on this extended PSF field to reduce the cost of physical operators, or to parameterize the operators by coordinates or deformation fields so that strong variance can be captured with fewer parameters. Finally, co-designing the hardware—for instance, masks or apertures that yield sparser or more localized PSFs—with the proposed reconstruction could further improve fidelity and make the joint system easier to invert.

**Limitation** While IFIN substantially improves reconstruction quality under severe blur and shift variance, it also has two practical limitations. First, the two-stream, multi-scale forward–inverse architecture with a shift-variant PSF field incurs higher computational and memory cost than simpler feed-forward or unrolled baselines, especially when using a large number of PSFs $k$. As analyzed, the runtime remains acceptable on modern GPUs, but real-time deployment on resource-constrained hardware will require further model compression and operator optimization. Second, our experiments focus on measurements acquired under moderate indoor lighting and a reasonably stable mask–sensor–scene geometry. As in other lensless systems, extreme high-dynamic-range scenes or strong external light can introduce large saturated regions on the sensor, and rapid or large geometry changes can invalidate the assumed forward model and the learned PSF field; in such regimes, information is physically missing and yet no single-exposure reconstructor, including IFIN and the baselines, can fully recover it. We view reducing the computational footprint and extending robustness to these challenging capture conditions as important directions for future work.

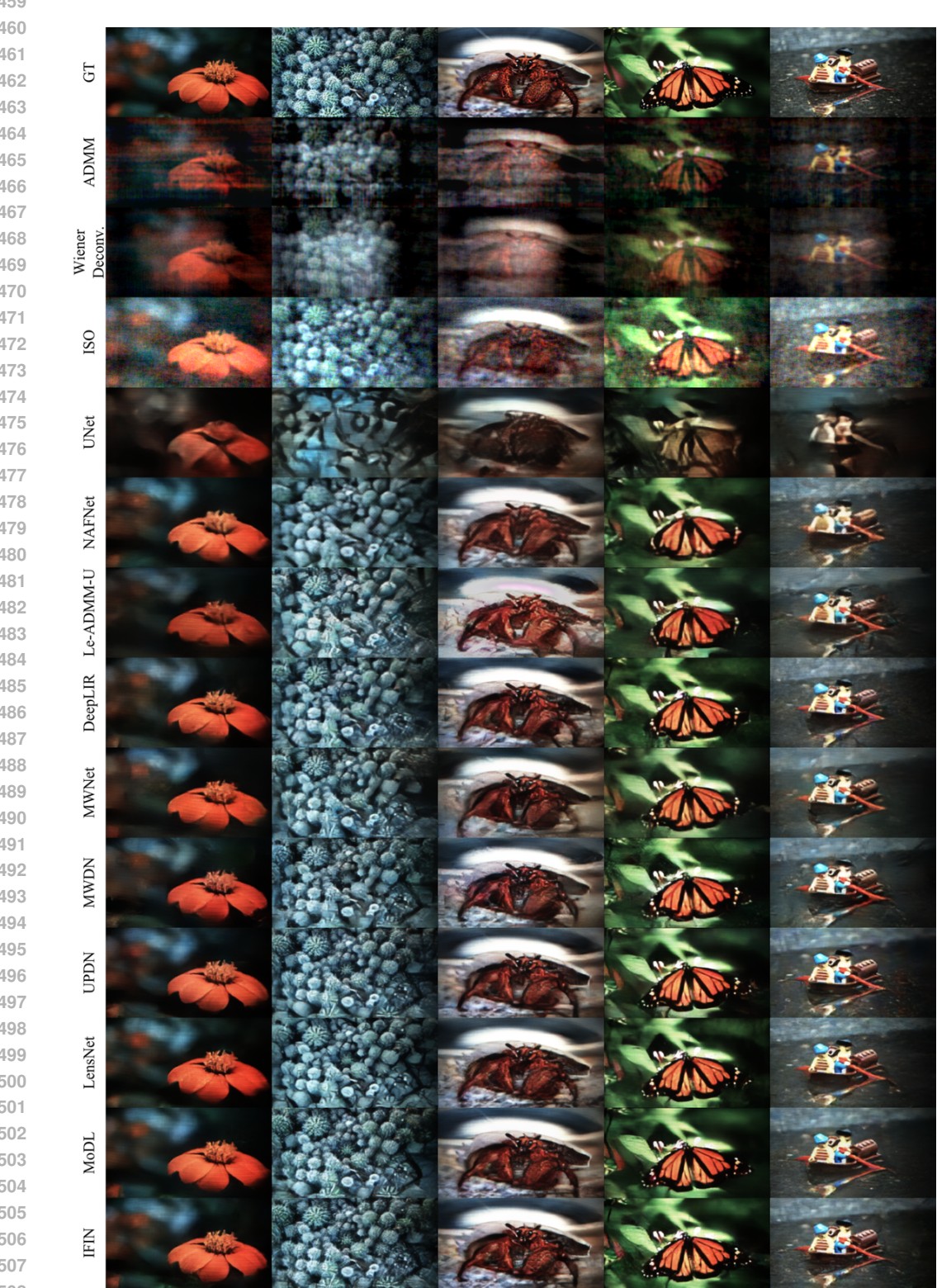

Figure 15: Additional qualitative comparison on the DiffuserCam dataset.

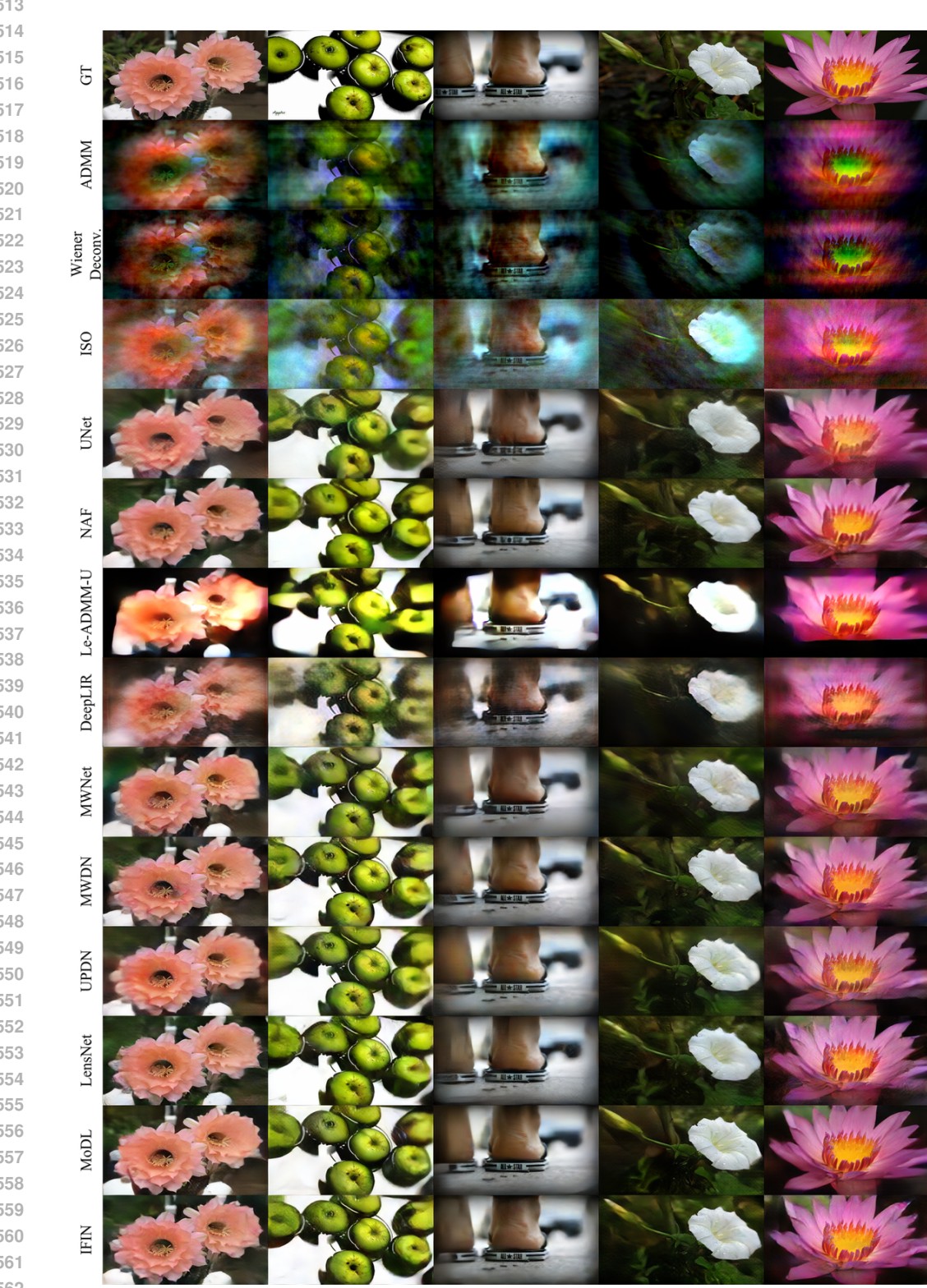

Figure 16: Additional qualitative comparison on the SV Lensless dataset.

