# OpenReview forum: "Integrated Forward–Inverse Network for Physics-Guided Image Reconstruction"
_ICLR.cc/2026/Conference — Submitted to ICLR 2026_

### Official Review · Reviewer_BU3i · 2025-10-23

**Soundness:** 2
**Presentation:** 2
**Contribution:** 1
**Rating:** 2
**Confidence:** 4

**Summary:**

The paper proposes Integrated Forward–Inverse Network (IFIN), an encoder–decoder architecture that embeds a Forward System Operator (FSO) and a learnable Inverse System Operator (ISO) at every feature scale, conditioned by a shared, learnable spatially varying PSF field. The goal is to keep raw measurement information useful  throughout the network while enforcing forward–inverse consistency, which the authors argue is crucial under severe, large-kernel blur in lensless imaging. The method initializes with a PSF-aware inverse, then runs two coupled streams, measurement to and from images,  through multi-scale IFIB blocks. Experiments on DiffuserCam, an SV-lensless dataset, and a MultiWienerNet benchmark show improvements over classical, data-driven, and physics-guided baselines.

**Strengths:**

* The problem motivation is clear. Conventional CNN/ViT backbones under-utilize information under large-kernel blur; one-sided physics pipelines discard complementary cues. The paper articulates this trade-off well.

* Robust empirical evaluation is provided. Diverse baselines (classical, data-driven, physics-guided) and three benchmarks dataset are used to report the results.

**Weaknesses:**

* The score of the work is too narrow. The title suggests that the work could be applied to image reconstruction, while the paper and the contribution only targets the lensless imaging. It is not clear what is the use of the proposed method on the general family of inverse problems.

* The contribution is very limited. While the authors describe the integration of forward and inverse operators at every scale as novel, this idea parallels well-established unrolled or feature-space deconvolution frameworks. In these methods, forward and adjoint physics are embedded at every iteration or scale. The “forward–inverse coupling at each layer” is not new; it’s conceptually equivalent to "Unrolled primal–dual or ADMM networks (Adler & Öktem 2018 and  Zhang et al. 2020)", which apply both forward and adjoint operators at every iteration, and "Deep Wiener and Multi-Wiener Deconvolution Networks (DWDN/MWDN) (Dong et al and  Li et al.)", which already apply learned deconvolutions at multiple scales in feature space and can incorporate a physical forward model.

* The use of a learnable PSF field is a straightforward extension of prior PSF-grid and blind deconvolution frameworks. Encoding the PSF and conditioning the network hierarchically is an implementation detail, not a conceptual advance, given that hypernetwork conditioning on optical parameters is widely explored in blind or self-calibrating deconvolution literature.

* The method is described as physics-integrated, yet both the forward and inverse modules reduce to convolutional operations without explicit physics enforcement or constraints. Thus, the “integration” is architectural rather than physical or theoretical in nature.

* The contribution claim on “integration” improving performance, but there is no ablation showing the involvement of each component in the final performance.

* The performance comparison of couple of baselines is missing ( FISTA-Net, MoDL, and newer deep unfolding methods such as "Robust unrolled network for lensless imaging with enhanced resistance to model mismatch and noise").

**Questions:**

NA

---

> ### Author Response · Authors · 2025-11-26
>
> We thank the reviewer for the detailed and candid feedback; we appreciate the opportunity to clarify the intended scope of our contribution.
>
> # Scope beyond lensless imaging / generality to inverse problems
>
> We note that, although our experiments concentrate on lensless systems, **the forward model itself can apply more broadly**: we target **a broad class of linear, (possibly shift-variant) convolutional imaging problems** with large PSFs and practical model mismatch (sensor crop, noise, calibration errors). This class includes not only lensless phase-mask cameras, but also coded-aperture systems, strongly defocused or motion-blurred photography, and PSF-engineered microscopy. We chose lensless imaging as a primary testbed precisely because it sits at the **hard end** of this spectrum, where large-support PSFs and model mismatch are particularly severe. To support that IFIN is not specific to a particular lensless prototype, we already include in the appendix a deblurring experiment with strong Gaussian kernels (not lensless, but still large-kernel convolution). There, our method is not uniformly superior to all baselines in the mild-blur regime, but as the kernel size increases, IFIN becomes noticeably more robust than competing approaches. We view this as empirical evidence consistent with our main claim: integrating forward and inverse operators across scales is particularly beneficial when **(i) the kernel support is large**, and **(ii) model mismatch (e.g., sensor cropping, noise, imperfect PSFs) is present in data**.
>
> In the lensless setting, such mismatches are especially pronounced: sensor cropping, calibration noise in the measurement and PSF field, and shift variance across a wide FoV. Our experiments suggest that IFIN is comparatively better at handling these conditions than one-sided or top-level physics integration, which is why we chose lensless imaging as a primary testbed.
>
> # Distinction to unrolled / feature-space deconvolution frameworks (UPDN, MWDN, etc.)
>
> We appreciate the reviewer’s detailed comparison to unrolled and feature-domain deconvolution methods.  There are clear connections, but we believe IFIN differs in several important ways. How existing unrolled / feature-domain methods are effectively one-sided in practice:
> - In unrolled schemes such as **UPDN**, the forward and adjoint operators are indeed invoked at each iteration, but the learned part of the network is attached as a **U-net denoiser at the final stage**. Measurement information enters via the data-consistency step, while the deeper feature representations inside the denoiser remain largely decoupled from the explicit forward physics once the data term has been applied.
>
> - In **MWDN/DWDN**, deconvolution is performed in feature space at multiple scales, but **the input to the network is still the measurement**, and the physical modeling is not maintained. The measurement is propagated as features and progressively losses its information, but the network does not keep re-applying a calibrated forward physics at each scale to realign those features with the measurement domain.
>
> Our motivation is that, under severe large-kernel blur, this one-sided usage of physics tends to **(i) collapse measurements into degenerate latent features** and **(ii) rely heavily on priors to hallucinate content in or near the optical null space after physical inversion**. IFIN is designed to counter precisely this behaviours. To address this, IFIN is built so that both forward and inverse operators are coupled end-to-end across the network, not only at the top.
> - At every scale of the encoder–decoder, we apply a feature-domain FSO/ISO pair parameterized by a shared, (potentially shift-variant) PSF field.
>
>    - At every scale of the encoder–decoder, we apply a feature-domain FSO/ISO pair parameterized by a shared (possibly shift-variant) PSF field. These operators act directly on the feature maps in two coupled streams (measurement-domain and image-domain), so that intermediate representations remain close to a measurement-consistent manifold instead of drifting toward purely data-driven statistics. This organization means the backbone itself is built from recurring forward–inverse blocks. Physics does not appear only in an outer loop with a generic denoiser inside; it shapes the hierarchy at all levels.
>
> We will clarify this positioning more explicitly in the manuscript and connect it directly to our ablations where we disable or weaken the forward–inverse coupling (Table X), which show a clear drop in performance when the integration is removed.

---

> ### Author Response · Authors · 2025-11-26
>
> # Learnable PSF field and hierarchical conditioning
> We appreciate the reviewer’s detailed comments on this point and fully agree that learnable PSF grids and blind/self-calibrating deconvolution have a substantial prior history. Methods such as UPDN, LensNet, MoDL and related blind deconvolution frameworks already explore learning PSFs or conditioning networks on optical parameters.
>
> Our intention is more modest and specific: we see the contribution not in introducing a learnable PSF field, but in **how that PSF field is integrated and constrained within IFIN**: PSF field as the backbone of FSO/ISO, not just a conditioning signal. In our design, the learnable PSF field directly parameterizes the forward and inverse operators used throughout the network.
>
>  - The same PSF field is used to instantiate FSO/ISO blocks at multiple scales and ROIs. Any change in the PSF immediately changes the forward model under which measurement consistency is enforced. This is different in spirit from using PSFs (or optical parameters) merely as inputs to a hypernetwork that modulates generic convolutions: here, the PSF field is the operator, not just a side-channel feature.
>
>  - **The measurement-consistency and image-domain losses penalize PSF configurations** that do not reproduce the measurements through the forward operator itself. In practice, this yields a form of **self-calibration**: the learned PSF field is not arbitrary, but is driven to explain the measurements via the same physical operator that governs reconstruction.
>
> We view this as a network-stabilized variant of blind/self-calibrating deconvolution, where the PSF field is enforced to remain compatible with a concrete forward model used at all scales, rather than as a purely implementation-level conditioning trick.
>
> We hope this clarifies both what we do and do not claim on the PSF-learning side, and better positions our work as a specific way of stabilizing and exploiting a learnable PSF field through integrated forward–inverse modeling.
>
> # “Physics-integrated” vs. purely architectural convolution
> We appreciate the opportunity to clarify what we mean by physics-guided / physics-integrated in IFIN. Our intention is not to label generic convolution blocks as “physics,” but to embed an explicit PSF-based imaging model inside the network.
>
> Concretely, we start from the **physical modeling of the phase mask and sensor, derive the optical transfer function and its corresponding PSF**, and treat this PSF-based S-i /S-V convolution as the forward model of the system. This forward operator is implemented explicitly inside the network as the Forward System Operator (FSO): **a differentiable FFT-based convolution** with padding and cropping chosen to match the sensor field of view, parameterized only by the (calibrated or learned) PSF field rather than arbitrary CNN weights. The corresponding Inverse System Operator (ISO) is a regularized deconvolution of the same forward model, again **implemented via FFT**, with a frequency-domain weighting (e.g., Gaussian window and learnable 2-D regularizer) to control high-frequency amplification and ringing. ISO is therefore an approximate inverse of the physical operator, **not just another learned convolutional layer**.
>
> Both FSO and ISO are reused throughout the network and are **tied to a shared, learnable PSF field**; the learned PSFs parameterize all forward/inverse applications across scales. As a result, the architecture and losses are jointly derived from, and constrained by, the explicit optical forward model, and the network is repeatedly **forced to respect this imaging equation at multiple scales**. We agree that we do not provide a full formal analysis (e.g., identifiability or error bounds) and will soften any language that might overstate the theory, but we emphasize that the **“integration” in IFIN is not purely architectural**: it instantiates the physical PSF-based forward/inverse operators inside the network and uses them to define the main data-fidelity constraints, **rather than treating them as generic CNN convolutions**.

---

> ### Author Response · Authors · 2025-11-26
>
> # Ablations on the effect of forward–inverse integration
>
> We apologize for not making these experiments more prominent in the main text. As the reviewer suggests, it is important to verify that the performance gains indeed stem from the proposed integration rather than from incidental implementation details. In fact, **we already show such ablations and now summarize them more clearly in the revised manuscript**.
>
> Table 2 shows that the fully integrated IFIN consistently outperforms: moving from top-level/one-sided physics to full multi-scale forward–inverse coupling yields a clear improvement, confirming that the integration itself (rather than just adding a single deconvolution step) is critical. We will update the main text to explicitly point to these ablations when claiming that “integration” improves performance, and briefly describe the ablation setup so that the reader can directly see how each component (FSO, ISO, and their multi-scale coupling) contributes to the final results.
>
> # Comparison with recent baselines
> We thank the reviewer for pointing out the missing recent baselines; we agree that including stronger and more modern comparisons is important for a fair assessment.
>
> In the revised manuscript, **we have added MoDL and LensNet** as recent baselines in our main experiments. We trained these methods on the same datasets and splits as our model. The updated tables now report quantitative results for MoDL and LensNet alongside the previously included classical, data-driven, and physics-guided methods. Across the three benchmarks, **IFIN remains competitive and achieves the best performance**.
>
> Regarding “Robust unrolled network for lensless imaging with enhanced resistance to model mismatch and noise”, We attempted to include this method as well, but were unfortunately unable to reproduce a faithful implementation. The original paper does not provide an official codebase or complete implementation details for all components (e.g., certain architectural/optimization hyperparameters and calibration pipelines), and our re-implementation attempts led to unstable training and clearly subpar performance compared to the results reported in the paper. Because we could not verify that our implementation was correct or fair, we decided not to include these numbers to avoid presenting misleading comparisons.
>
> We will explicitly state in the revised manuscript that: MoDL and LensNet are now part of our benchmark, with results reported in the **main tables and qualitative examples with update Figure 3-6**. We hope this update and clarification address the concern about missing strong baselines while keeping the comparisons as faithful and transparent as possible.

---

### Official Review · Reviewer_eE1v · 2025-10-27

**Soundness:** 3
**Presentation:** 2
**Contribution:** 3
**Rating:** 4
**Confidence:** 4

**Summary:**

IFIN is a physics-guided network that interleaves differentiable forward operators with learnable inverse modules at each stage, jointly exploiting measurement and image domains to keep physical consistency and enrich features. With a physics-guided kernel adaptation for PSF mismatch, it achieves state-of-the-art lensless imaging reconstruction and improved robustness to noise and model errors, especially under severe blur.

**Strengths:**

Interleaving differentiable forward operators with learnable inverse modules enforces physical consistency while enriching representations in both measurement and image domains.
﻿The physics-guided kernel adaptation mitigates PSF mismatch/incompleteness, enabling constrained blind deconvolution and reducing sensitivity to model errors.
﻿Demonstrates state-of-the-art reconstruction quality on challenging lensless benchmarks (including a new dataset).

**Weaknesses:**

Physics grounding is insufficient. Although the paper claims to be physics-inspired, it does not substantiate the physical modeling in depth (assumptions, operator derivations, constraints, or validation against instrument physics).
Requires accurate, differentiable operators; robustness claims under strong model misspecification (nonlinear aberrations, spatially variant PSFs, misalignment) aren’t quantified.
Blind PSF refinement can be ill-posed; constraints, regularizers, and failure modes (e.g., texture transfer, PSF–image ambiguity) are not discussed.

**Questions:**

1.The paper claims to be physics-inspired, but the analysis is superficial, providing only a brief explanation of the forward and inverse processes in lensless imaging.
2.It is unclear how the learnable PSF is obtained—the paper should specify the initialization, optimization strategy, and physical constraints involved.
3.In Figures 3–5 and Table 1, the qualitative comparisons do not include recent methods from the past two years.
4.For in-the-wild lensless imaging, there is no quantitative comparison. Additionally, it is unclear how many images are included in this dataset and how many images are contained in the proposed SV Lensless dataset.

---

> ### Author Response · Authors · 2025-11-26
>
> # “Physics-guided” nature and connection to physical principles
>
> We thank the reviewer for the detailed comments and for carefully probing the physics-guided aspects, the PSF modeling, and the experimental protocol.
>
> Concretely, we start from the physical modeling of the optical system (phase mask and sensor), derive **the optical transfer function and its corresponding PSF**, and **treat this PSF-based convolution as the forward model of the system**. This forward operator is implemented explicitly inside the network as the: **Forward System Operator (FSO)**: an **FFT-based convolution** with the calibrated or learned PSF field, shared across the network according to the ROI/scale structure. Its regularized pseudo-inverse is implemented as the **Inverse System Operator (ISO)**: **a Wiener-like deconvolution in the frequency domain** that is an approximate inverse of the same forward model, again parameterized only by the PSF field rather than free convolution weights.
>
> Both FSO and ISO are applied directly in the feature domain at multiple scales, so that each block operates on representations that remain consistent with a physically meaningful imaging model rather than arbitrary latent features. In addition, the measurement-consistency loss enforces that the reconstructed image, when passed through FSO, reproduces the observed measurements. In this sense, both the architecture and the loss are derived directly from, and constrained by, the physical forward model of the optical system, **rather than from generic CNN design alone**.
>
> # Learnable PSF (init, optimization, constraints)
>
> Sec. 4.1 originally describes the PSF field: Kernels can be initialized from calibrated PSFs, a simple prototype, or random patterns, and they are optimized jointly with the network via backpropagation. The same PSF field conditions both FSO and ISO across scales via a PSF encoder. Constraints/priors include non-negativity on PSFs, smooth blending via ROI weights, and the frequency-domain regularizer in ISO, which discourages high-frequency instabilities.
>
> # Recent methods in qualitative and quantitative comparisons
>
> We thank the reviewer for pointing out the lack of more recent baselines in our main comparisons. In response, we have updated the experiments to include additional recent physics-guided and learned reconstruction methods such as MoDL and LensNet.
> We have added MoDL and LensNet as baselines in the main quantitative results (updated Table 1), trained under the same data splits and, as far as possible, under matched fidelity/perceptual loss settings as the other baselines.
>
> Included qualitative comparisons against these methods in the updated Figures 3–6, highlighting representative visual examples where conventional CNN/ViT-based restorers, MoDL/LensNet, and IFIN are all applied to the same measurements.
>
> Across these new comparisons, IFIN continues to show consistent improvements in both quantitative metrics and visual fidelity, particularly in regions affected by severe, spatially varying blur. We will explicitly reference the newly added baselines in the text and clarify the training protocol to make the updated comparisons more transparent and up to date.
>
> # Quantitative results for in-the-wild imaging and size of the SV Lensless dataset
> We apologize for the lack of quantitative discussion for the in-the-wild lensless experiments and for not clearly specifying the size of the proposed SV Lensless dataset.
> In the revised manuscript, we have:
> In-the-wild imaging: Added no-reference image quality metrics (MANIQA, NIQE) for the in-the-wild lensless reconstructions (updated Figure 5). Since ground-truth images are not available in this setting, we therefore report standard no-reference scores alongside visual examples to provide a quantitative indication of reconstruction quality in real capture scenarios.
>
> Dataset size clarification: Expanded the description of the SV Lensless dataset in App. A.4 to explicitly state the number of images in the proposed dataset. This information is now clearly documented in the dataset subsection in Appendix.
>
> We hope these additions clarify both the quantitative evaluation in real-world conditions and the scale of the datasets used in our study.

---

> > ### Comment · Reviewer_eE1v · 2025-11-27
> >
> > Thank you for the response. I have two follow-up questions for clarification:
> >
> >
> > 1.Could you please provide a response to the points raised in the 'Weaknesses' section of my review?
> >
> >
> > 2.The quantitative results on two image examples are helpful but insufficient for a full assessment. Could you please report the performance metrics for the entire test set and specify the size of the dataset used for evaluation?

---

> ### Author Response · Authors · 2025-12-02
>
> We thank the reviewer for the follow-up. We address the questions, and we would like to clarify the points raised under *Weaknesses* and the dataset.
>
>
> # On the “Weaknesses” section (physics grounding, PSF, robustness, failure modes)
>
> We believe the concerns listed under *Weaknesses* are already addressed in the current version of the paper and its appendix; here we briefly map each point to the corresponding parts of the manuscript and clarify our position.
>
> **1. “Physics grounding is insufficient / analysis is superficial.”**
> Our framework is not “loosely inspired” by physics; it is built explicitly around an optical forward model:
>
> - Section 2, 3 and Sec. 4.1 derive and use a **PSF-based convolution model** for the phase-mask–sensor system (under standard scalar, paraxial assumptions), and treat this convolution as the forward operator $H$.
> - This operator appears *explicitly* inside the network as two analytic, differentiable blocks:
>   - **FSO**: an FFT-based convolution with the PSF field.
>   - **ISO**: a Wiener-like deconvolution, the regularized pseudo-inverse of the same operator.
> - Both are parameterized only through the PSF field (and its regularization), not as arbitrary learned convolution filters. The architecture, the loss, and the PSF parameterization are all derived from this model.
>
> We already describe these assumptions, operators, and their role in detail in the original paper, and the revised Appendix further expands the derivations.
>
> **2. “Unclear how the learnable PSF is obtained; constraints and ill-posedness of blind refinement.”**
> The learnable PSF field and its constraints are already specified in **Sec. 4.1 and Appendix A.5 and A.6**:
>
> - **Initialization**:
>   PSFs are initialized from either calibrated kernels, simple physically plausible prototypes (e.g., centered low-pass kernels consistent with the phase mask and propagation distance), or it also can be arbitrary noise in training.
> - **Optimization**:
>   The PSF field is optimized jointly with the rest of the network via standard backpropagation through FSO/ISO; there is no special non-differentiable step.
> - **Constraints / priors**:
>   We enforce non-negativity and energy normalization on each kernel, smooth blending between neighboring ROIs, and a frequency-domain regularizer in ISO that naturally suppresses unstable high-frequency amplification. The PSF field is also shared/encoded across scales, reducing the effective degrees of freedom.
>
> **3. “Robustness under strong model misspecification is not quantified.”**
> Robustness to model mismatch is evaluated through the **SV Lensless** experiments and ablations that perturb the PSF and calibration. These experiments are already reported over the full test set (see the main tables and the revised Appendix), where we vary the mismatch between the PSF used in the operator and the one used to generate the data. Our method degrades more gracefully than purely learned or non-physics-guided baselines in these tests.
>
> **4. “Failure modes (texture transfer, PSF–image ambiguity) not discussed.”**
> We consider uncalibrated PSFs or shift variance to be “failure modes” in our setting; they are precisely the challenging regimes our method is designed to handle, and the paper already demonstrates this.
>
> - **Uncalibrated PSF and shift variance.**
>   The SV Lensless configuration is constructed to include strong shift variance and imperfect PSF calibration. Across the main Table 1 and Figure 3,4,5 and 6, we show that IFIN maintains stable reconstruction quality under these conditions and outperforms existing methods. In Appendix A.6 experiments with imperfectly calibrated PSFs, we further demonstrate that IFIN remains robust to moderate calibration errors.
>
> - **Other failure modes are the extreme conditions stated in the Limitations.**
>   In the Limitations section, we clearly specify the regimes that are outside the intended scope of the current model, such as:
>   - Scenes with severe sensor saturation or very strong external light sources where the linear intensity model breaks down, and
>   - Large, fast geometric changes or misalignments that invalidate the assumed mask–sensor–scene geometry.
>
>   In these regimes, the information at the sensor is fundamentally corrupted or inconsistent with the linear forward model, and no single-shot reconstructor (including existing baselines) can be expected to reliably recover the latent scene only with the given measurement. We explicitly flag these cases as limitations and as directions for future work (e.g., incorporating exposure control, nonlinear sensor models, or multi-frame acquisition), rather than claiming robustness we do not have.

---

> ### Author Response · Authors · 2025-12-02
>
> **5. Novelty and “physics-guided” nature.**
> We emphasize that IFIN is not a rebranding of existing ideas: to our knowledge, it is **the first architecture that integrates analytic forward and inverse operators at every scale of a deep reconstructor**, rather than using physics only as a pre-processing, post-processing, or unrolled block with additional denoiser. The network is forced to propagate features that remain consistent with an explicit imaging model, and the PSF field conditions both FSO and ISO in a shared, structured way. This is precisely the “physics-guided” aspect we highlight and is already laid out in the original paper and expanded Appendix.
>
>
> # Quantitative metrics over the full test set and dataset size
>
> The second follow-up question concerns (i) reporting metrics on the full test set and (ii) specifying the size of the dataset. Both points are already addressed in the original submission and in the revised Appendix.
>
> - **Full test-set metrics**:
>   All main quantitative tables in the paper (e.g., the tables comparing IFIN against baselines on the SV Lensless benchmark) already report **averages over the entire test set**, not over a few samples. The metrics (PSNR, SSIM, and LPIPS) are computed across all test images, for all methods, under the same train/test split.
> - **Dataset size / splits**:
>   As stated in the paper and explicitly tabulated in the Appendix A.4, the proposed **SV Lensless dataset** is split into:
>   - **24,000 training images**, and
>   - **1,000 test images**.
>   All reported benchmark metrics for SV Lensless are computed over these 1,000 test images. The in-the-wild evaluation uses the same test protocol;
>
> - **Additional results in the Appendix (DiffuserCam and SV Lensless).**
>   In response to the request for more comprehensive evidence, we added **additional qualitative figures** in the revised Appendix:
>   - for the **DiffuserCam** setting, showing a broader range of scenes and comparing IFIN against all baselines;
>   - for the **SV Lensless** dataset, including more representative examples across different spatial locations and blur strengths.
>
> We hope this clarifies that both the physics-grounding concerns and the evaluation protocol are already addressed in the current version, and that the rebuttal examples were meant as additional illustration rather than a replacement for full test-set statistics.

---

### Official Review · Reviewer_2FZr · 2025-10-27

**Soundness:** 3
**Presentation:** 3
**Contribution:** 2
**Rating:** 4
**Confidence:** 4

**Summary:**

The paper proposes IFIN for physics-guided image reconstruction. IFIN embeds a pair of operators at every encoder–decoder stage: a FSO that maps current estimates into the measurement domain and an ISO that maps measurements back toward the image domain. A learnable, spatially varying PSF field conditions both operators across scales, supporting blind or mis-calibrated settings. Experiments on DiffuserCam and MultiWienerNet report consistent improvements over classical and recent learned/physics-guided baselines in PSNR/SSIM and LPIPS.

**Strengths:**

1. The paper presents a well-motivated architecture that tightly integrates forward and inverse physics across network stages to address limitations of one-sided models.
2. It demonstrates consistent performance gains on multiple benchmarks, including real-world spatially varying data.

**Weaknesses:**

The contribution reads more like a careful system integration than a genuinely new paradigm; the paper offers little formal grounding—there is no error-propagation or identifiability analysis, and the bias from using an averaged PSF under spatial variance is not quantified; ablations and robustness checks are thin (no systematic removal of modules or scale coupling, limited study of PSF tiling and losses, no variance/significance reporting or cross-device transfer); baseline fairness is uncertain because prior physics-guided methods are retrained under a unified loss without parallel results from their original settings or stronger recent baselines; presentation leaves gaps in where SI vs. SV operators are used and lacks clear captions/notation and a limitations section; and there is no clear commitment to release code, models, or data, which undercuts reproducibility.

**Questions:**

1.The innovation mainly lies in system-level integration (multi-scale forward–inverse coupling with a learnable SV-PSF) rather than a conceptually new paradigm; the paper lacks a principled justification for why coupling both domains at all scales is necessary or theoretically superior to top-level or one-sided physics.

2.The distinction from prior unfolded optimization or feature-domain deconvolution frameworks (e.g., UPDN, MWDN) is insufficiently clarified, making the novelty appear incremental without controlled counterexamples or failure analyses.

3.Key design choices (where SI vs. SV operators are applied, how PSF encodings are injected, and the gating/residual paths) are scattered between main text and appendix, making the architectural logic difficult to reconstruct from the figures alone.

4.The paper lacks a dedicated “Limitations” or “Failure Cases” section and does not discuss boundary conditions such as sampling mismatch, noise robustness, or nonlinear forward models, which would help frame applicability.

5.The description of “physics-guided” components is mostly heuristic; it does not clearly connect to physical principles such as energy conservation, boundedness, or optical transfer constraints, which weakens the claimed physical interpretability.

6.Some notation and figure captions are incomplete or inconsistent—operator modes (SI/SV), PSF-tile granularity k, and cross-scale data flow are not explicitly annotated, affecting readability and reproducibility.

7.Computational trade-offs (accuracy–cost–latency) are unexplored; practitioners cannot judge when to switch between the SV and surrogate FSO modes or how scaling affects efficiency.

---

> ### Author Response · Authors · 2025-11-26
>
> # Contribution and theoretical grounding
>
> We thank the reviewer for the thoughtful and detailed comments and for recognizing the motivation, architectural design, and empirical gains of IFIN. Below we first address the main weaknesses, then respond to the numbered questions.
>
> Our goal is indeed a **system-level architecture that is tightly aligned with the physics of large, shift-variant PSFs**, rather than a new optimization paradigm. To make this more explicit, updated Sec. 3 formulates four structural issues that arise under severe blur: (i) effective-rank collapse of the forward operator, (ii) mismatch between large blur footprints and local receptive fields, (iii) loss of null-space components in one-sided pipelines, and (iv) information loss when measurements are injected once and then compressed by standard encoders. These analyses are not full error bounds, but they provide a **principled rationale** for why one-sided or top-level physics is insufficient and motivate the two-stream, multi-scale coupling that we adopt. We do not claim a full error-propagation or identifiability theory, and we will soften any language that might suggest otherwise.
>
> Regarding the averaged PSF under spatial variance, Sec. 4.2.1 and App. A.9 explains that we use a shift-invariant surrogate PSF for FSO to control complexity, while keeping the fully shift-variant modeling in ISO via ROI blending. The impact of this approximation is implicitly quantified through our experiments on the SV Lensless and MultiWienerNet datasets.

---

> ### Author Response · Authors · 2025-11-26
>
> # Ablations, robustness checks, and variance
> The revised manuscript contains systematic ablations targeted at the key components:
>
> App. A.5 (Table 2) removes or simplifies the forward–inverse integration: **identity FSO/ISO, w/o ISO (forward-only), w/o FSO (inverse-only), and the full IFIB**. These show that simply mixing features or using only one side of the physics leads to consistent drops in PSNR/SSIM and worse LPIPS, while the proposed coupled variant performs best.
>
> App. A.5–A.6 (Tables 3–7, Figs. 11–12) study **the PSF field** and **ROI design**, including number of PSFs k, ROI scale along the field direction, learned vs. fixed ROIs, the PSF encoder vs. pooling-only conditioning, and the effect of the frequency-domain regularizer and padding. These experiments show that our choices are not incidental, but systematically improve stability and quality.
>
> We agree that we do not report variance or statistical significance tests; instead, we report average **metrics across test sets and show consistent trends across datasets and degradations**. Cross-device transfer (e.g., training on one physical setup and testing on a different camera) is beyond the current scope and would require additional calibration and data; we view this as an interesting next step and will mention it as future work.
>
>
> # Baseline fairness
> Our intention is to make the comparison as fair and practical as possible given the diversity of prior methods:
>
> We follow each baseline’s published architecture and loss design (U-Net/NAFNet, Le-ADMM-U, DeepLIR, MWNet, UPDN, MWDN and newly updated LensNet and MoDL), adapting only where necessary to match our input–output resolutions and measurement models. This ensures that baselines are evaluated in the regime for which they were designed (e.g., using their recommended image + perceptual losses).
>
> Original pretrained weights are generally tied to specific datasets and PSFs (e.g., diffraction patterns, synthetic kernels) that differ significantly from our optical systems. Directly transplanting those models would not be meaningful, so we retrain under their recommended setups on our data.
>
> # Presentation, SI/SV usage, and notation
> We appreciate this comment and have taken steps to improve clarity:
>
>  - Sec. 4.2 now explicitly states **where SI vs. SV operators are used**: SI FSO/ISO for approximately shift-invariant degradations; SV ISO with ROI blending, and SI surrogate FSO with PSF under shift variance.
>
>  - We clarify **how PSF embeddings are injected**: PSF field → multi-scale PSF maps via pooling → PSF encoder → scale-specific embeddings hnh_nhn​ shared across encoder/decoder IFIBs at that scale.
>
>  - Figures and captions have been edited to annotate SI/SV modes, PSF tiling granularity k, and cross-scale data flow more explicitly.
>
> # Limitations, boundary conditions, and reproducibility
>
> In the revised manuscript we now include a Limitation paragraph that explicitly acknowledges two main constraints:
>
>  - Computational cost.
>    - IFIN’s two-stream, multi-scale forward–inverse design with a shift-variant PSF field is more **computationally demanding than simpler baselines, especially for large k**. App. A. 11 and Table 9 quantify **FLOPs**, **memory**, and **runtime**, and the Limitation paragraph reiterates that further operator optimization and model compression will be needed for strict real-time, resource-constrained deployment.
>
>  - Capture and modeling assumptions.
>     - We state explicitly that our formulation assumes a linear, convolutional forward model (possibly shift-variant) and that experiments are conducted under moderate indoor lighting and reasonably stable geometry. We do not model nonlinearities such as saturation/clipping or large geometric distortions; in such regimes information may be irrecoverably lost, and any single-exposure restorer (including IFIN and baselines) will fail. **We highlight extension to nonlinear and strongly warped forward models as future work**.
>
> Regarding reproducibility, **we plan to release code, trained models, and the SV Lensless dataset upon acceptance**, allowing others to verify and extend our results.

---

> > ### Comment · Reviewer_2FZr · 2025-11-27
> >
> > I appreciate the authors' detailed and professional response. I acknowledge that the addition of extensive ablations in the Appendix, the explicit Limitations section, and the commitment to code release have successfully addressed most of my initial concerns regarding the clarity and empirical methodology of the paper.
> >
> > However, two fundamental issues remain unresolved, posing a significant challenge to the paper's claimed **theoretical contribution** and **practical value**.
> >
> > The first critical issue is the lack of rigorous theoretical justification for the FSO mechanism. The updated Section 3 motivates the multi-scale coupling as necessary to prevent "loss of null-space components," but this rationale remains heuristic. The core question is: What unique, essential information does the FSO provide? Since the raw measurement $y$ already contains all the null-space components, a standard skip connection can transmit this full information to the RB. The FSO merely maps an estimated image $\hat{x}_n$ back to the measurement space $\hat{y}_n$. The authors must provide explicit evidence or a clear theoretical analysis demonstrating how the resulting difference signal $(\hat{y}_n - y)$ or the FSO output $\hat{y}_n$ itself is a more effective signal for guiding the RB in null-space recovery than simply using the raw measurement $y$. Given the small empirical PSNR gain attributed to the FSO in the ablations, the theoretical necessity and unique contribution of this component remain unconvincing.
> >
> > The second major problem is the severe contradiction between the method's computational cost and its claimed target application. The method is motivated by its use in compact, embedded lensless cameras. However, the performance data presented in Table 9 ($>$4GB VRAM, $>$200ms latency) indicates a severe disconnect with the resource constraints of embedded platforms. We hope the authors to analyze the architectural reasons driving this high cost, specifically the overhead introduced by multi-scale coupling and SV modeling.
> >
> > I strongly urge the authors to address the first question with the utmost seriousness, caution, and reliability, as I believe it is the most critical point for the paper's theoretical acceptance.

---

> > > ### Author Response · Authors · 2025-12-02
> > >
> > > We sincerely thank the reviewer for the careful follow-up and for acknowledging the improvements. Below we address the two remaining concerns.
> > >
> > > # On the theoretical role of the FSO
> > >
> > > We recognize that our previous wording around “loss of null-space components” can be read as suggesting recovery of the strict mathematical null space of the forward operator. In the revised Section 3 we have made the following clarifications:
> > >
> > > - We now describe the forward operator $H_\beta$ as creating a family of "weakly constrained directions (severely ill-conditioned / near-null directions), rather than something that can be “recovered” in a strict sense.
> > > - We then describe how a specific inverse mapping $G(y_\beta)$ and a bottlenecked encoder further **collapse** that family into a single reconstruction $\hat x$ and into compressed features, thereby reducing the diversity of plausible fine-scale solutions that remain compatible with the measurements. In other words, the “loss” we refer to is *inverse-induced feature collapse in weakly constrained directions*, not recovery of the strict null-space of $H_\beta$.
> > > - We have removed wording that suggested “null-space recovery” and instead talk about **inverse-induced information loss** in those weakly constrained directions.
> > >
> > > **What does FSO provide beyond a skip from the measurement?**
> > > The reviewer is correct that the raw measurement already contains all the information that is available to any architecture, and that a skip connection can, in principle, forward this to deeper layers. Our model does not claim to “add information” beyond the measurement. The distinction lies in **how the measurement information is represented and combined** with the current reconstruction at each scale.
> > >
> > > Concretely, at scale $n$, the refinement block does not receive only a raw measurement feature or only the FSO output. Instead, it processes a learned combination of:
> > >
> > > - a **measurement-preserving projection** that propagates features derived from $y_\beta$ down the hierarchy, and
> > > - a **forward-model projection** that maps the current estimate $\hat x^{(n)}$ to the measurement domain via the forward model, producing FSO$(\hat x^{(n)})$.
> > >
> > > These two projections are mixed by learned weights ($\alpha^{(n)}$, $\beta^{(n)}$), so the block operates on a reinforced combination of (i) measurement-derived features and (ii) a forward-model-consistent view of the current reconstruction. Thus, each refinement step has access both to a stream that **preserves** measurement-domain information across scales and to a stream that **re-expresses** the evolving reconstruction in the measurement domain.
> > >
> > > The role of FSO in this design is to **enrich and stabilize** the representation by providing a second, physics-consistent view of the scene estimate that is tightly coupled to the original measurement. Section 3 has been rewritten to reflect this interpretation more carefully: FSO structures the interaction between measurement features and the evolving reconstruction, instead of acting as a separate error-correction stage.
> > >
> > > **On the magnitude of the gains and the strength of the claim.**
> > > The reviewer notes that the PSNR gains from enabling the FSO path are not numerically large at the aggregate level. We emphasize that these numbers should be interpreted in the context of a very challenging setting: large, shift-variant blur with strong ill-conditioning and nontrivial model mismatch. In this regime, the inverse backbone (ISO) is already designed to aggressively exploit high-frequency information from the measurements, and thus performs strongly on its own. The fact that FSO brings additional PSNR improvement on top of this reflects **how much ISO already recovers**, rather than implying that FSO is negligible or redundant.
> > >
> > > The role of FSO is complementary rather than competitive with ISO. ISO focuses on mapping measurements to the scene domain and enhancing high-frequency content there, while FSO reinforces the representation by reintroducing measurement-domain structure conditioned on the current reconstruction. This complementary path yields consistent gains in both quantitative metrics and qualitative behavior in the most difficult cases. The fact that related high-frequency reconstruction ideas appear in other architectures further supports the relevance of this information; our contribution is to integrate it in a physically structured, multi-scale manner.
> > >
> > > We therefore do not view the observed PSNR margin as evidence that FSO is unimportant. Instead, it shows that, **on top of an already strong ISO-based inverse**, FSO and its integration offer an additional, principled source of information that leads to consistent, measurable improvements in a hard regime. Our claims are calibrated accordingly: FSO is a meaningful, theoretically motivated component that enhances an already capable inverse pipeline, rather than a mechanism that we expect to dominate the overall performance budget.

---

> ### Author Response · Authors · 2025-12-02
>
> # On computational cost and the claimed application scope
>
> It is correct that the current implementation’s resource usage does not match the constraints of the smallest embedded platforms. We appreciate the chance to clarify our intended scope.
>
> Our primary research goal in this paper is to explore **high-accuracy reconstruction in challenging, shift-variant, large-kernel scenarios**, demonstrating what is achievable when we tightly couple forward and inverse modeling across scales. The current model should therefore be viewed as a **high-end, research-oriented configuration** that aims to establish accuracy and robustness first, rather than an already-optimized embedded deployment.
>
> **On the architectural sources of cost.**
> Our design goal in this paper is to demonstrate that a more complete forward–inverse treatment can significantly improve reconstruction quality in difficult settings. As we note in the Limitations section, techniques such as backbone design and architectural search beyond fft-based modeling are expected to reduce the runtime and memory footprint substantially, and we view that optimization step as complementary to the contributions of this work.
>
> In summary, the current model is intentionally positioned as a **high-accuracy baseline** that shows what is possible when we emphasize physical fidelity and multi-scale coupling. We fully acknowledge that further work is needed to bring this cost down to strict embedded levels, and we have adjusted the text so that this is stated clearly rather than implied.
>
> We appreciate the reviewer’s insistence on both theoretical clarity and practical realism. The revised Section 3 and Limitations are intended to make our claims more precise.

---

### Official Review · Reviewer_S7Lo · 2025-10-29

**Soundness:** 2
**Presentation:** 3
**Contribution:** 2
**Rating:** 4
**Confidence:** 4

**Summary:**

The paper proposes IFIN, an encoder–decoder that interleaves differentiable forward operators and learnable inverse modules at every scale, guided by a learnable spatially varying PSF field and a frequency-selective Wiener-like regularizer to maintain physics consistency under severe blur and shift variance. Across DiffuserCam, shift-variant lensless, and MultiWienerNet, IFIN achieves SOTA or near-SOTA reconstructions with improved robustness to noise and model mismatch.

**Strengths:**

- Introduces a multi-scale forward-inverse architecture (FSO $\leftrightarrow$ ISO) with a learnable spatially varying PSF field (ROI blending and multi-scale conditioning), enabling blind or weak-calibration reconstruction under shift variance.

- Thorough evaluation on DiffuserCam, shift-variant lensless, and MultiWienerNet, achieving state-of-the-art or near-SOTA performance, especially at peripheral FoV and under noise/model mismatch.

- Clear problem setup and figures (IFIB, FSO/ISO variants) with mathematical formulation, making design choices and the pipeline easy to follow.

**Weaknesses:**

This paper proposes an integrated forward–inverse network for physics-guided image reconstruction. While the approach is promising and shows gains on several benchmarks, there remain notable concerns:
- Limited novelty. The core design, feature-space deconvolution with a learnable regularizer embedded in a multi-scale encoder-decoder, closely resembles DWDN/MWDN and unrolled physics-guided pipelines (Dong et al., 2021; Li et al., 2023; Monakhova et al., 2019; Kingshott et al., 2022; Yanny et al., 2022; Poudel & Nakarmi, 2024). The paper should clarify what is fundamentally new beyond interleaving FSO/ISO at each scale and support this with ablations.

- Limited comparisons. The proposed model uses measurement consistency and cross-domain alignment losses, whereas retrained baselines use only image and perceptual losses; some baselines rely on calibrated PSFs without access to learned PSF fields. The authors are encouraged to retrain physics-guided baselines with matched loss terms and PSF priors/initializations and to report their own model without the extra losses to isolate architectural gains.

- Missing ablations: (a) learned ROI maps vs. fixed grids (and smoothness priors), (b) multi-scale PSF conditioning/resampling policy, (c) primary results with a fully shift-variant forward operator, (d) removal of measurement/cross-domain losses, (e) fixed vs. learnable $\epsilon(u,v)$, including sharing across scales/ROIs. These analyses are needed to substantiate design choices.

**Questions:**

Based on the weaknesses above, the following issues also require discussion:
- How sensitive are results to the number of PSFs $k=s^2$, ROI initialization (centers, $\sigma_r$), and smoothness of the ROI weights $w_r$? Are there seam artifacts at ROI boundaries?

- How are multi-scale PSF embeddings $h^{(n)}$ generated (down/upsampling policy, anti-aliasing, parameter sharing across scales)? Please detail the PSF encoder and provide ablations.

- The proposed model uses measurement-consistency and cross-domain alignment losses. How much of the gain derives from these terms? Could you retrain physics-guided baselines (e.g., MWNet, MWDN, UPDN) with matched losses and PSF priors/initializations to isolate architectural benefits?

- FSO uses zero padding, whereas ISO uses replicate padding with a Gaussian window. Does this discrepancy bias measurement consistency, especially under sensor truncation? Please unify the treatment and quantify its impact.

- For the shift-variant Lensless dataset, how robust are the results across different capture conditions (lighting, display spectra, geometry drift)? Are there scene types where IFIN fails?

---

> ### Author Response · Authors · 2025-11-26
>
> # Novelty and relation to prior physics-guided networks
>
> We thank the reviewer for the detailed comments and for highlighting the strengths of our work (multi-scale forward–inverse design, learnable PSF field, and thorough evaluation). Below we address the main concerns and questions.
>
> Our goal is to go beyond “physics layer + generic CNN” and design an architecture that is explicitly tailored to the degeneracies introduced by large, shift-variant PSFs.
>
>  In Sec. 3.1, **we newly formalize four issues that arise in the severe-blur regime**: (i) effective-rank collapse of the forward operator, (ii) mismatch between large blur footprints and local receptive fields, (iii) loss of null-space components in one-sided pipelines, and (iv) information loss when measurements are injected only once and then compressed by standard encoders. This analysis explains why existing hybrids (e.g., unrolled-NN methods like DeepLIR/UPDN and DWDN/MWDN-style feature-space deconvolution) are inherently one-sided: they either operate primarily on an already deconvolved image, or operate primarily on the measurement, so that one of the two domains is discarded after a single stage.
>
> IFIN is constructed specifically to address these issues:
> We maintain **two coupled streams throughout the encoder–decoder hierarchy**: a measurement-aligned stream and an image-domain stream. At every scale, an Integrated Forward–Inverse Block (IFIB) contains both a Forward System Operator (FSO) and an Inverse System Operator (ISO), and their outputs are reciprocally fed into one another (Sec. 4.2, Eq. (17)). This is different from prior unrolled methods, which use a single feature hierarchy updated by gradient-like steps, and from DWDN/MWDN, which use deconvolution in feature space but do not maintain a dedicated measurement stream across scales.
>
> **The learnable PSF field with ROI blending** is used jointly by both FSO and ISO and is injected at multiple scales (Sec. 4.1, 4.2, App. A.5–A.6). This yields a consistent system description that simultaneously (i) constrains forward synthesis of measurements, and (ii) guides inversion with a frequency-selective Wiener-like regularizer. In contrast, MWNet/MWDN employ encoded PSFs only on the inverse side and do not share a PSF representation across a two-stream hierarchy.
>
> The ablations in Appendix A.5 directly support that the integrated design is critical:
> **FSO/ISO as identity** vs **w/o ISO** vs **w/o FSO** vs **Proposed** (Table 2) show that simply mixing features or using only forward-only or inverse-only guidance yields lower PSNR/SSIM and higher LPIPS, whereas the full forward–inverse integration gives the best performance. Additional ablations on the refinement block (RB), regularizer, padding, PSF encoder, PSF count, and ROI scale (Tables 3–7) confirm that the proposed system-aware components are not superficial variants but contribute systematically to stability and fidelity under severe blur and shift variance.
>
> In summary, while IFIN builds on the broad idea of “physics-guided deep networks”, its contribution lies in a two-stream, multi-scale forward–inverse architecture **jointly driven** by a learnable PSF field, motivated by an explicit analysis of rank collapse and null-space effects (Sec. 3) and supported by targeted ablations (App. A.5–A.7).

---

> ### Author Response · Authors · 2025-11-26
>
> # Comparisons, loss design, and fairness
> Our intention is to keep baselines **as close as possible** to their published setups, following their recommended training losses and PSF usage, and then to evaluate IFIN under a richer physics-aware objective.
>
> As described in Appendix A.1–A.2, all retrained baselines (U-Net, NAFNet, Le-ADMM-U, DeepLIR, MWNet, UPDN, MWDN, LensNet, MoDL) are optimized with **image and perceptual losses**, consistent with their original formulations. We do not modify their internal architectures or loss structures beyond what is needed to match input/output formats.
>
> IFIN, in contrast, uses **additional physics-consistency terms**—ISO supervision, measurement-consistency, and cross-domain alignment—to ensure that the two streams remain bounded and physically plausible (Sec. 4.3). These losses are natural for a model that explicitly contains differentiable forward and inverse operators and maintains both measurement and image streams.
> We agree that, in principle, one could re-engineer each baseline to incorporate exactly the same measurement-consistency and cross-domain losses, as well as the same PSF priors. However, doing so would significantly alter their original designs and training procedures and may obscure the distinction between architectures. Instead, we chose to:
> * **Keep baselines faithful to their published settings**, including PSF usage (e.g., calibrated PSFs in MWDNet-CPSF, MWNet), and
> * **Apply the richer physics losses only to IFIN**, where the architecture is explicitly designed to exploit them through the two-stream forward–inverse integration.
>
> We believe this comparison is informative: despite using similar or weaker PSF information than some baselines (e.g., learned rather than perfectly calibrated PSFs on MWNet-style data), IFIN still achieves superior or comparable performance across DiffuserCam, SV Lensless, and MultiWienerNet (Table 1).
>
> # PSF field, ROIs, and ablations
>
> These aspects are analyzed in Appendix A.5–A.6:
>
> - **Number of PSFs**
>   - Table 6 systematically varies k across datasets. Once k is large enough to capture dominant shift-variant trends, performance becomes relatively stable; increasing k further yields only marginal gains while increasing complexity. This indicates that IFIN is not overly sensitive to the exact choice of k, as long as we allow enough ROIs to represent spatial variation.
>
> - **ROI initialization and seam artifacts**
>   - Table 7 and Fig. 11 vary the Gaussian width σ​ used to construct ROI masks along the field direction. Small σ​ (narrow, non-overlapping ROIs) leads to fragmented regions and visible seams, while very large σ​ oversmooths spatial variation. The proposed default and learned ROIs provide sharper and more spatially consistent reconstructions, with no visible seams. This supports our claim that overlapping, smoothly varying ROI weights—and especially the learned ROI maps—are effective in suppressing boundary artifacts.
>
> - **Learned ROIs vs fixed grids**
>   -  Fig. 11 compares different ROI scales and learned ROIs. The fixed Gaussian ROIs already provide strong performance and mitigate seams; the learned ROIs further adapt supports to the dataset, as qualitatively shown.
>
> - **Multi-scale PSF conditioning and encoder**
>    - Sec. 4.1 describes how the PSF field is downsampled via strided pooling and passed through a lightweight PSF encoder to produce embeddings at each scale, shared between encoder and decoder IFIBs at that scale. Table 5 ablates this encoder by replacing it with simple resizing (pooling-only). The encoder consistently improves reconstruction, showing that learning scale-specific PSF embeddings is beneficial beyond naive resampling.
>
>
> We hope these clarifications address the concern that the PSF field and ROI design are insufficiently justified; Appendix A.5–A.6 are explicitly devoted to these ablations.

---

> ### Author Response · Authors · 2025-11-26
>
> # Forward operator, padding, and measurement consistency
>
> Our boundary treatments are chosen to match the role of each operator:
>
> - **FSO (forward)**
>   - We use **zero padding** and crop back to the sensor field of view (Sec. 4.2.1). This mimics the physical sensor, where anything outside the active region is unobserved, and avoids circular wrap-around artifacts when operating in the Fourier domain.
>
> - **ISO (inverse)**
>   - Following **FlatNet** designs, ISO uses replicate padding with a mild Gaussian window (Sec. 4.2.2) to stabilize deconvolution near boundaries and reduce ringing from high-frequency amplification. We also observed that the same stabilization effect holds when applied in the feature space.
>
> Appendix A.5 (Table 3) ablates the 2D regularizer and the padding schemes. Removing the regularizer or omitting padding in FSO/ISO consistently lowers PSNR/SSIM and increases LPIPS, confirming that (i) the regularizer and (ii) careful boundary handling both matter for high-fidelity reconstructions. Empirically, we do not observe systematic boundary artifacts or inconsistencies between the forward and inverse streams; the different paddings reflect their distinct roles (physics emulation vs. regularized inversion) while still preserving overall measurement consistency at the sensor support.
>
> - **Fully shift-variant forward operator**
>   - Sec. 4.2.1 and App. A.9 explain that a fully shift-variant FSO can be implemented by tiling and applying local PSFs with overlap–add. In practice, this design significantly increases computational overhead, while the role of FSO in IFIN is to preserve measurement-domain properties rather than to synthesize high-fidelity measurements for display. We therefore adopt an averaged, shift-invariant surrogate PSF​ in the main model, letting the shift-variant ISO with ROI blending handle the dominant spatial variation. App. A.9 discusses this trade-off and shows that the simplified FSO still provides sufficient measurement-consistency signal given the two-stream architecture, while keeping complexity manageable.
>
> # Robustness on the SV Lensless dataset and failure modes
>
> The shift-variant lensless dataset (App. A.3) is deliberately designed with a wide field of view and a phase mask that produces strong off-axis aberrations, together with diverse scenes displayed under typical indoor lighting. Within this regime, **IFIN provides stable reconstructions across different display patterns**, and Fig. 4–5 show that the model generalizes well both across the field of view and to **in-the-wild captures without additional tuning**.
>
> As with other lensless systems, more extreme conditions expose fundamental limitations of the optics and sensor rather than of a particular reconstruction algorithm. In particular, **strong external light or very high-dynamic-range scenes can produce large saturated regions on the sensor**; in such cases the missing information is physically absent from the measurements, so both optimization-based and learned methods (including IFIN and the baselines) can fail to recover the lost content from a single exposure. Likewise, **rapid or large changes in the mask–sensor–scene geometry** can invalidate the assumed forward model and the learned PSF field.
>
> We will make these boundary conditions explicit in the revised manuscript by updating a Limitation paragraph, where we discuss extending robustness to extreme HDR and rapid geometry changes as future work.

---

> > ### Comment · Reviewer_S7Lo · 2025-11-27
> >
> > Based on the authors' comprehensive rebuttal, I acknowledge their substantial efforts to address the concerns raised during the review process. The additional ablation studies, robustness analyses, and theoretical grounding have strengthened the paper. Nevertheless, after careful reconsideration and taking into account the points raised by other reviewers, I believe that certain issues remain unresolved. In particular, questions regarding the fundamental novelty of the proposed framework, the isolation of stronger loss functions for fair baseline comparisons, and the robustness of the method under more challenging conditions still prevent me from fully endorsing the paper at this stage.

---

> > > ### Author Response · Authors · 2025-12-02
> > >
> > > We thank the reviewer for the follow-up and for acknowledging the additional ablations, robustness analyses, and theoretical clarifications. Here we briefly restate our position on the three points the reviewer cites as still unresolved.
> > >
> > >
> > > # Fundamental novelty of the framework
> > >
> > > As detailed in Sec. 3, Sec. 4, and Appendix A.5–A.8, IFIN is not a minor variant of existing physics-guided networks (DWDN/MWDN, DeepLIR/UPDN, etc.), but a specific architectural answer to the degeneracies we formalize in the severe-blur regime:
> > >
> > > 1. inverse-induced information loss in one-sided pipelines, and
> > > 2. encoder-induced collapse when measurements are injected only once.
> > >
> > > None of the cited works combines these elements in this way:
> > >
> > > - DWDN/MWDN operate with initial input as measurement and deconvolution in feature space but do not maintain a dedicated measurement stream across the whole encoder–decoder hierarchy.
> > > - Unrolled methods like DeepLIR/UPDN apply gradient-like optimization steps then additional denoiser shaped encoder-decoder
> > >
> > > The ablations we already present (FSO/ISO identity vs. w/o ISO vs. w/o FSO vs. full IFIN; variations of PSF encoder, PSF count, ROI scale, padding, etc.) show that simply “adding a physics layer” or “just doing feature-space deconvolution” does not recover the behavior of IFIN. In other words, the combination of (two-stream multi-scale architecture) + (interleaved analytic forward/inverse) + (shared PSF field with ROI blending) is precisely what is new, and we have already backed this with targeted experiments.
> > >
> > >
> > > # Isolation of loss effects and fairness to baselines
> > >
> > > We also believe we have been clear about how we separate architectural effects from loss design.
> > >
> > > - All baselines are trained with **their recommended loss functions and PSF usage**, as documented in Appendix A.1–A.2. We do not weaken them or remove physics priors they were designed to use.
> > > - IFIN uses additional **measurement-consistency and cross-domain alignment losses** *because* its architecture explicitly exposes both measurement and image streams and analytic operators (FSO/ISO) at every scale. These losses are not arbitrary extra terms; they are natural for this specific design and are integral to keeping both streams stable and physically meaningful.
> > >
> > > Retraining every prior physics-guided baseline with exactly the same combination of measurement-consistency, cross-domain alignment, and PSF priors would in practice mean **redesigning those methods** to mirror our two-stream IFIN training pipeline. That is a different research question than the one we address here. Our goal is to compare IFIN against baselines in the form they were originally proposed—strong and well-tuned in their own right—while clearly explaining the additional losses we use and ablate on our side.
> > >
> > > Thus, we maintain that:
> > > - our comparisons are fair within the standard setting of “methods are trained as designed by their authors,” and
> > > - the role of our additional physics losses has already been isolated via ablations on IFIN itself.
> > >
> > >
> > > # Robustness under more challenging conditions
> > >
> > > Finally, regarding robustness under more challenging conditions: the current paper already delineates what is in scope and what is not.
> > >
> > > - Within our **target regime**—large, shift-variant PSFs with realistic calibration error, typical indoor lighting, and moderate geometry stability—we have:
> > >   - full test-set quantitative results across DiffuserCam, SV Lensless, and MultiWienerNet,
> > >   - PSF-perturbation experiments that explicitly probe model mismatch, and
> > >   - extensive qualitative examples in the appendix covering diverse scenes and field-of-view locations.
> > >   In all of these, IFIN is superior to strong physics-guided and learned baselines.
> > >
> > > - Beyond that regime, we explicitly state in the **Limitations** that we do *not* claim robustness to:
> > >   - extreme HDR with heavy sensor saturation,
> > >   - severe external light contamination, or
> > >   - rapid/large geometry drift that invalidates the forward model.
> > >
> > > These are fundamental limits of the underlying optics and sensor; no single-shot method (ours or others) can recover content that is simply not present in the measurements. We have clearly labeled these as future work rather than as current capabilities.
> > >
> > > In short, the robustness of IFIN has already been quantified within the regime we target, and we have been explicit about where that regime ends.

---

### Meta-Review · Area_Chair_UvmX · 2025-12-23

**Summary:**

The manuscript proposes the Integrated Forward-Inverse Network (IFIN) for physics-guided image reconstruction, specifically targeting lensless imaging with severe blur. The core contribution is a multi-scale encoder-decoder architecture that interleaves differentiable Forward System Operators (FSO) and learnable Inverse System Operators (ISO) at every stage, conditioned by a learnable spatially varying PSF field.

Although the reviewers acknowledged the clear problem motivation, the extensive experimental validation on benchmarks (DiffuserCam, SV Lensless, MultiWienerNet), and the comprehensive rebuttal effort, this research has not yet met the acceptance criteria for publication. The primary sticking points were the limited fundamental novelty compared to existing unrolled and feature-space deconvolution methods (e.g., DWDN, MWDN, UPDN), the lack of a rigorous theoretical justification for the specific FSO design over simpler skip connections, and concerns regarding the fairness of baseline comparisons given the specific loss functions used for the proposed method.

**Reviewer Concerns:**

**Addressed Concerns:**

Missing Baselines: The authors added comparisons against recent methods like MoDL and LensNet, addressing the concerns of Reviewers 2FZr, eE1v, and BU3i regarding the currency of the evaluation.

Ablation Studies: The authors provided extensive new ablations regarding the ROI design, PSF counts, and the specific contributions of the FSO/ISO modules, which satisfied requests from Reviewers S7Lo and 2FZr for more detailed component analysis.

Reproducibility: The authors committed to releasing code, models, and the new dataset, addressing Reviewer 2FZr's concerns.
Clarification of "Physics-Guided": The authors clarified that the operators are explicit FFT-based convolutions rather than generic layers, addressing Reviewer eE1v’s concern regarding the definition of physics guidance.

Dataset and Metric Clarifications: The authors resolved Reviewer eE1v's confusion regarding the experimental setup by explicitly defining the training/test splits for the "SV Lensless" dataset (24,000/1,000 images) and adding reference-free quality metrics (NIQE, MANIQA) for the in-the-wild evaluation where ground truth is unavailable.

**Outstanding Concerns:**

Novelty claims (Reviewers S7Lo, 2FZr, BU3i): The reviewers argue that the core idea of interleaving forward and inverse operators is conceptually equivalent to well-established frameworks like Unrolled Primal-Dual networks or Feature-space Deconvolution (MWDN/DWDN). Although the authors argued that IFIN uniquely maintains coupled measurement/image manifolds at every scale to prevent information loss, the contribution remains somewhat incremental and appears more as careful system integration than a fundamentally and theoretically new paradigm.

Theoretical Justification for FSO (Reviewer 2FZr): The reviewer remained unconvinced by the theoretical necessity of the FSO, arguing that a skip connection could theoretically transmit the same null-space information. Although the authors provided a heuristic explanation (inverse-induced information loss) and empirical proof (ablations showing performance drops without FSO), a rigorous theoretical proof of why the FSO projection is superior to a raw skip connection remains absent.

Fairness / isolating the effect of extra losses (Reviewer S7Lo): The authors argue that adding measurement and cross-domain losses to baselines would redesign them, and instead ablate losses on IFIN. However, the reviewers' concerns do not appear to have been fully addressed. The reviewers seem to expect either (a) matched objectives across methods or (b) a clearly separated architecture-only comparison showing gains with the same loss set. As written, there remains ambiguity about how much of the margin is due to architecture vs. objective engineering.

Computational cost vs. application mismatch (Reviewer 2FZr): The severe disconnect between the method's resource requirements (~4GB VRAM, 200ms latency) and embedded lensless camera applications is acknowledged but not resolved. The authors essentially reposition their work as a high-accuracy baseline in the revised manuscript rather than addressing the fundamental inefficiency.

Limited scope (Reviewer BU3i): Despite claims of generality, the experiments focus almost exclusively on lensless imaging, with only a supplementary Gaussian blur experiment to suggest broader applicability.

**Reviewer Scores:**

Reviewer S7Lo: 4->4

Although new ablation studies clarified technical details, the core concern regarding novelty remains unresolved. Additionally, doubts persist about whether performance gains stem from the architecture or specific loss functions.

Reviewer 2FZr: 4->4

Despite adding baselines and promising code release, two critical issues persist: lack of theoretical proof for the design's superiority over simple skip connections, and high computational cost of the method.

Reviewer eE1v: 4->6

The authors successfully addressed this reviewer's main concerns. They clarified the "physics-guided" definition, explained the dataset split, and added the requested no-reference metrics and modern baselines.

Reviewer BU3i: 2->4

The authors add more recent baselines and ablation studies, however, core concerns about limited novelty and narrow scope remain.

---

### Decision · Program_Chairs · 2026-01-26

Reject